# FARM: Enhancing Molecular Representations with Functional Group Awareness

## Abstract

We introduce **F**unctional Group-**A**ware **R**epresentations for Small **M**olecules (**FARM**), a novel foundation model designed to bridge the gap between SMILES (a linear string representation of molecular structures), natural language, and molecular graphs. The key innovation of FARM lies in its functional group (FG) annotation at the atomic level, which enables both **FG-enhanced SMILES** and **FG graphs**: SMILES are enriched with FG information to specify which functional group each atom belongs to, while the FG graph captures the molecular backbone by showing how the functional groups are connected. This tokenization not only injects chemical knowledge into SMILES but also expands the chemical lexicon, effectively bridging the gap between SMILES and natural language in terms of vocabulary size, making the sequences more suitable for Transformer-based models. FARM then learns molecules from two complementary perspectives to fully encode both functional and structural information. Masked language modeling on FG-enhanced SMILES captures atom-level features enriched with FG context, while graph neural networks encode higher-level molecular topology by representing how FGs are connected. Contrastive learning then aligns these two views into a unified embedding, ensuring that atom-level details and FG-level structure are jointly represented. This dual-level modeling is central to FARM's ability to predict molecular properties accurately. We rigorously evaluate FARM on the MoleculeNet dataset, achieving state-of-the-art performance on 8 out of 13 tasks, and further validate its generalization on the photostability dataset for quantum mechanics properties. These results highlight FARM's potential to improve molecular representation learning, demonstrate its strong transfer learning capabilities across drug discovery and material design domains, and make way for broad applications in pharmaceutical research and functional materials development. The code is available at: https://anonymous.4open.science/r/farm_molrep-291C.

## 1 Introduction

Artificial intelligence (AI) has emerged as a transformative tool in accelerating scientific discovery, particularly in drug development and material design. In these domains, it is increasingly employed for tasks such as molecular property prediction, drug–target interaction prediction, quantitative structure–activity relationship (QSAR) modeling, and the discovery of novel materials with tailored electronic, mechanical, or optical properties. (Chen et al., 2016; Wen et al., 2017; Shen and Nicolaou, 2019; Walters and Barzilay, 2020; Achary, 2020; Wang et al., 2022a; Edwards et al., 2022; Zhang et al., 2023a; Nguyen et al., 2024; Edwards et al., 2024b;a). However, one of the central challenges in this field is the scarcity of large labeled datasets required for traditional supervised learning methods. This has shifted the focus towards self-supervised pre-trained models that can extract meaningful patterns from vast amounts of unlabeled molecular data (Shen and Nicolaou, 2019). As a result, the development of robust foundation models for molecular representations is now more critical than ever.

Despite significant advancements in other domains, such as natural language processing (NLP) and computer vision, there is still no dominant foundation model tailored to molecular representation in drug discovery (Zhang et al., 2023b). This paper begins to address this pressing gap by introducing an innovative *functional group*

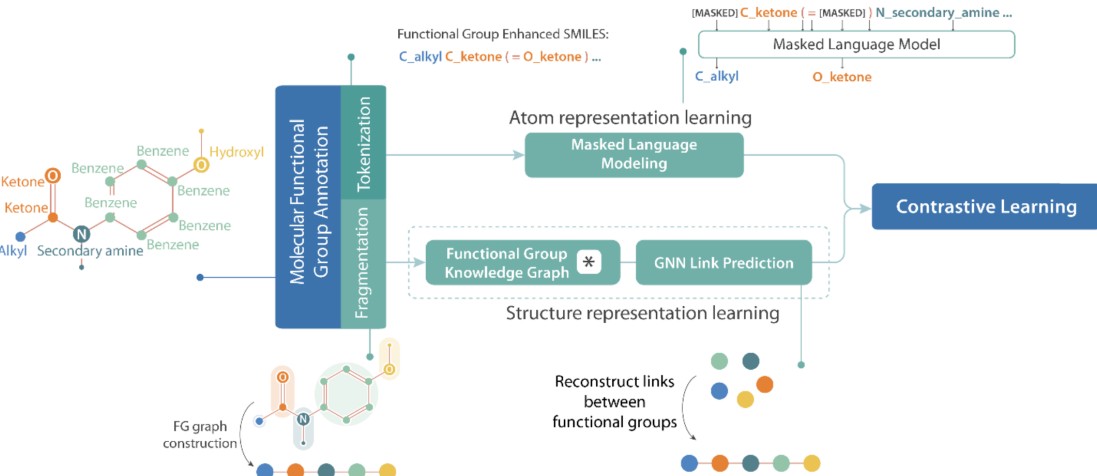

Figure 1: FARM's molecular representation learning framework. Given a molecule as input, we apply FG-aware tokenization to its SMILES representation and FG-based fragmentation to its molecular graph. Each resulting representation is then processed through a self-supervised learning objective. Finally, the two views are aligned using contrastive learning to make the final embedding robust by capturing both atom-level information from the SMILES and structural information from the graph.

*(FG)-aware* approach, in which classical FGs are identified according to established chemical principles, while also considering additional chemically meaningful substructures that strongly influence molecular properties. Functional groups play a central role in determining the chemical and biological behavior of molecules, and even small changes can lead to significant differences in activity. For instance, salicylic acid and aspirin share the same molecular backbone but differ by a single functional group: the hydroxyl group in salicylic acid is replaced by an acetyl group in aspirin. This subtle modification dramatically alters their properties and biological activities, as the acetyl group in aspirin enables irreversible inhibition of cyclooxygenase (COX) enzymes, producing blood-thinning effects that help prevent heart attacks and strokes, effects not observed with salicylic acid. By rigorously grounding tokenization in **chemical rules**, our approach aims to more effectively capture these critical molecular properties and functions.

In the strictest chemical sense, functional groups refer to well-defined moieties such as hydroxyl, carbonyl, or amino groups. In this work, we adopt a broader interpretation of FGs to include not only these classical groups but also ring systems and fused ring systems, which, although not always labeled as functional groups in textbooks, often play critical roles in molecular recognition, stability, and activity.

The key innovation of our approach lies in how molecules are tokenized. In SMILES representations, we enrich detail by annotating each atom with its FG, while in molecular graph representations, we abstract detail to emphasize the molecular backbone, namely which FGs are present and how they are connected. This dual strategy allows molecular embeddings to more faithfully capture functional information, enabling more accurate prediction of molecular properties and functions.

For example, instead of representing every oxygen atom with the same token ("O"), we assign different tokens based on their role in functional groups, such as "O_ketone" or "O_hydroxyl"(Figure 1). Traditional sequence-based models typically treat atoms only by type and ignore the functional groups they belong to, even though a single atom can play very different roles depending on its chemical context. This is analogous to natural language, where a word like "bank" can mean either a river bank or a financial institution. By specifying an atom's functional group, our approach enriches SMILES with chemically contextual information. We refer to this representation as **FG-enhanced SMILES**, which enriches molecular sequences with chemically meaningful context and expands the vocabulary beyond the limited set of atom symbols in standard SMILES.

This vocabulary expansion is particularly important because standard SMILES have a very limited and repetitive vocabulary, essentially atom and bond types, with the majority being C, O, and N. Training models

on such a restricted corpus often leads to rapid convergence, but the representations capture little about functional or property-related patterns. As a result, downstream performance can suffer and even exhibit negative transfer (Xia et al., 2023). By enriching SMILES with FG annotations, we **enlarge the vocabulary** and bring it closer in spirit to natural language, the domain for which Transformer models are designed. This richer representation helps bridge the gap between SMILES and natural language, thereby mitigating negative transfer and improving the model's ability to learn molecular properties and functions.

While FG-enhanced SMILES captures fine-grained chemical details, sequential representations like SMILES often struggle to represent the overall molecular backbone– the higher level arrangement of functional groups that defines molecular topology. To address this, we construct a **FG graph**, where each node corresponds to a functional group and edges encode their connectivity. For example, a ketone group (C=O) is represented as a single FG node rather than two separate atoms, producing a high-level molecular backbone. Each FG node is assigned a chemically informed embedding, learned from a **FG knowledge graph** that captures both the intrinsic properties of each FG and their relationships with other FGs across molecules. These embeddings are then used to initialize a graph neural network (GNN), which is trained via a **link prediction** task to reconstruct connections between FGs. By predicting which FGs are likely to be connected, the GNN learns the molecule's high-level structure, relationships that are often implicit in SMILES representations.

Finally, **contrastive learning** is employed to align the two views: the **FG-enhanced SMILES** and the **FG graph**, encouraging the model to learn unified representations that are sensitive to chemically meaningful structures. Figure 1 and Figure 2a present an overview of our molecular representation learning model.

In summary, our key contributions include:

- **FG-aware tokenization and fragmentation:** We introduce an FG annotation algorithm that labels each atom in a molecule with the FG to which it belongs. The FG-annotated molecular graph is then used for FG-aware tokenization and fragmentation, producing both FG-enhanced SMILES and FG graphs. This approach enriches each atom with chemical context, bridges the gap between SMILES and natural language through a FG–aware vocabulary, and still captures higher-level structural information in the graph.

- **Structural representation learning:** We construct a FG knowledge graph to obtain robust FG embeddings, and learn molecular structure by predicting the links between FGs, thereby capturing the high-level structural organization of molecules.

- **Atom-feature and structural representation integration:** We combine a masked language model to learn atom-level features with GNNs to capture structural information, and align these two views using contrastive learning.

- **Robustness in downstream tasks:** FARM demonstrates strong transfer learning capabilities, the core goal of pretrained models, outperforming other methods in 8 out of 13 tasks from MoleculeNet, which spans physiology, biophysics, physical chemistry, and quantum physics. Beyond MoleculeNet, FARM further shows robust generalization on quantum mechanics datasets, including the photostability dataset.

## 2 Related Work

**Functional group-Aware Molecular Representations** In recent years, there has been growing recognition that incorporating functional group (FG) information into molecular representations can significantly enhance model performance in downstream tasks. Approaches in this domain can be broadly classified into two categories: those leveraging language models (LMs) (Li et al., 2023; Xia et al., 2023) and those utilizing graph neural networks (GNNs) (Zhang et al., 2020; 2021; Yu and Gao, 2022; Yang et al., 2022; Wang et al., 2023; Wu et al., 2023; Han et al., 2023; Fang et al., 2023). Within these categories, methods either employ rule-based functional group detection, relying on predefined chemical rules to identify FGs, or adopt unsupervised strategies that infer substructures or motifs from the data. Regardless of the approach, these models consistently demonstrate that enriching molecular representations with functional group information leads to improved performance across a wide range of molecular property prediction tasks (Fang et al., 2023; Han et al., 2023; Wang et al., 2023). This underlines the importance of functional group awareness in achieving accurate and generalizable molecular representations. More recently, SCAGE (Qiao et al.) further validates this direction by incorporating functional group prediction as one of four pretraining objectives in a

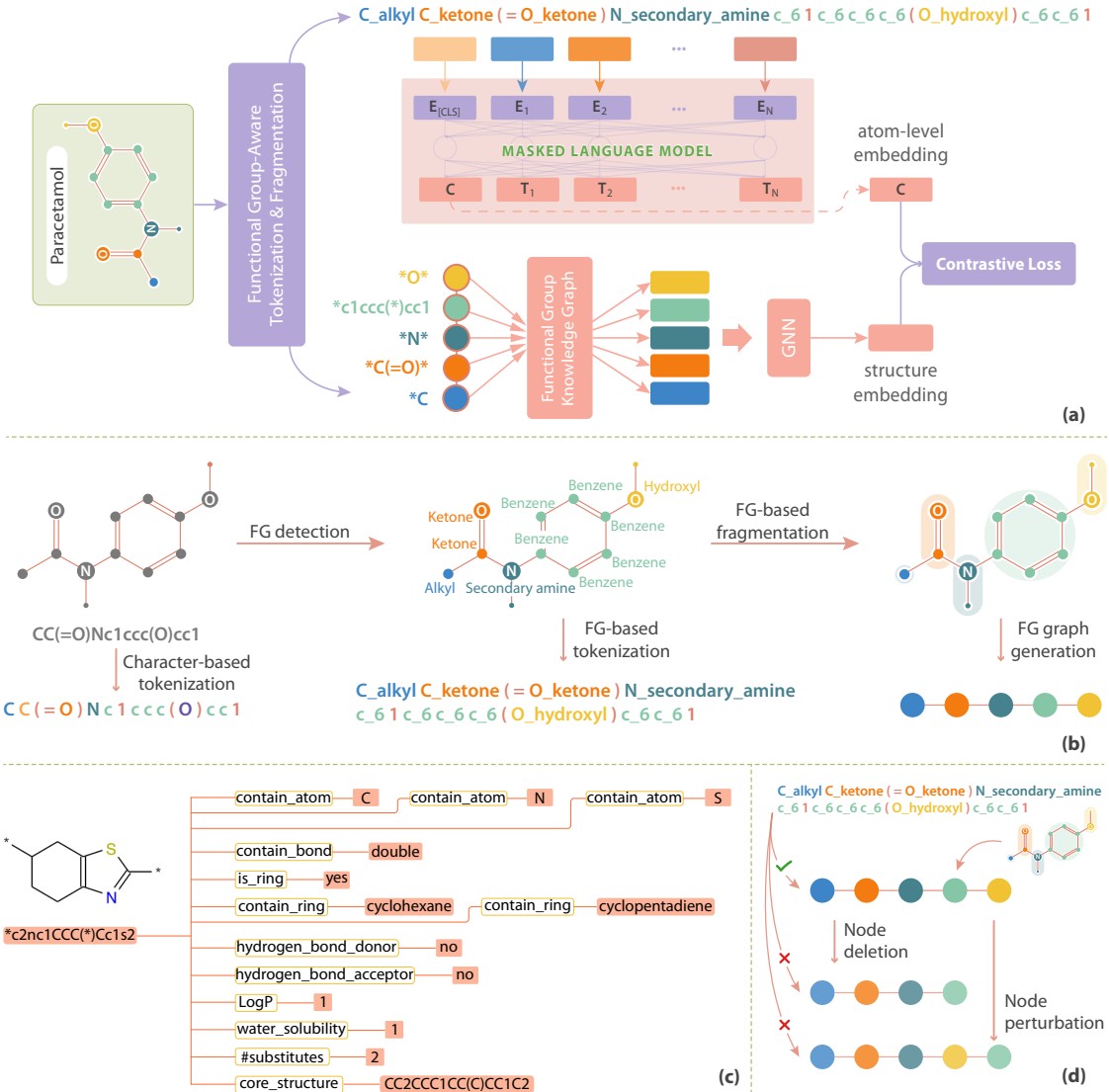

Figure 2: (a) FARM's molecular representation learning model architecture. (b) Functional group-aware tokenization and fragmentation algorithm. (c) Snapshot of the functional group knowledge graph. (d) Generation of negative samples for contrastive learning.

self-conformation-aware graph transformer, demonstrating that explicitly modeling functional groups at the atomic level leads to improved generalization across diverse molecular property prediction tasks.

In studies such as (Zhang et al., 2021; Han et al., 2023; Chen et al., 2024; Yang et al., 2022), the BRICS algorithm (Degen et al., 2008) is employed to fragment molecules, with some extending the approach through additional rules to achieve finer-grained segmentation. However, this method focuses on general reaction-based bond breaking, rather than targeting functional group-specific characteristics. Additionally, while BRICS groups atoms into fragments, it does not explicitly identify or label functional groups, limiting its utility in tasks that require functional groups-specific information. Other works (Li et al., 2023; Chen et al., 2024) leverage RDKit (Landrum, 2010) for functional group detection. While RDKit is effective in identifying common functional groups, its capabilities are limited to well-known groups and do not extend to detecting more complex functional groups, such as ring systems that often dominate molecular datasets and play a critical role in molecular function. In this work, we present a rule-based functional group detection algorithm that accurately identifies 101 common groups and all ring-containing ones. Unlike frequent subgraph mining

or motif-based methods, our approach is explicitly grounded in chemical principles, ensuring reliable and chemically meaningful detection aligned with expert definitions.

**Contrastive Learning-Based Molecular Representations**   Contrastive learning, a self-supervised learning technique, aims to learn representations by maximizing agreement between positive pairs while distinguishing them from negative samples. In the context of molecular representation, Wang et al. (2022b) applies this technique by augmenting each molecular graph to create slightly different versions, which are then treated as negative examples to enhance the learning process. In Pinheiro et al. (2022), the authors use a molecular graph encoder and a SMILES encoder to encode both molecular graphs and SMILES, employing contrastive loss to maximize the agreement between these embeddings. This method enriches SMILES with topology information and the molecular graph with sequence context, making the final embeddings of molecules more robust. In Zhang et al. (2020), the authors minimize the distance between the representation of a molecule and the representations of its constituent substructures. Collectively, these works highlight the versatility of contrastive learning in unifying diverse molecular representations, leading to improved downstream performance in molecular tasks. Our method extends this approach by using contrastive learning to maximize the agreement between FG-enhanced SMILES representations and FG graphs, thereby enhancing the alignment between sequence-based and graph-based representations, ensuring a more comprehensive and chemically informed embedding.

---

**Algorithm 1** Functional group annotation for molecular graph

---

**Require:** Molecular graph $G$, functional group definitions $FG\_list$ (ordered largest to smallest)
 1: **for** each atom $v$ in $G$ **do**
 2:    **if** $v$ does not have a functional group assigned **then**
 3:       $FG \leftarrow$ largest functional group in $FG\_list$ that matches $v$ and its neighborhood
 4:       $matched\_atoms \leftarrow$ atoms involved in the match
 5:       **for** each atom $a$ in $matched\_atoms$ **do**
 6:          $a$.SetProp("FunctionalGroup", $FG$.name)
 7:       **end for**
 8:    **end if**
 9: **end for**

---

# 3 Methodology

We present our approach for enhancing molecular representation learning through FG-enhanced SMILES learning with Transformers, FG graph backbone modeling, and dual-perspective contrastive learning, jointly capturing atom-level chemical context and FG-level molecular structure. Section 3.1 introduces FG-aware tokenization and fragmentation, which injects detailed chemical context into both sequence and graph representations. Section 3.2 delves into the masked atom prediction task, a self-supervised technique that enables the model to learn atom-level representations. In Section 3.3, we describe our method for capturing the core molecular structure. Finally, Section 3.4 presents contrastive learning, which aligns FG-enhanced SMILES strings with FG graph embeddings to achieve a comprehensive, unified molecular representation.

## 3.1 Functional Group-Aware Tokenization and Fragmentation

We propose an FG-aware tokenization and fragmentation method that incorporates detailed functional group information into molecular representations, designed for both SMILES and graph-based models. Our approach defines a set of functional groups and uses an algorithm to detect them within the molecular graph, annotating each atom with its corresponding FG. The algorithm traverses the molecular graph, evaluating each atom based on criteria such as atom type, neighboring atom types and bonds, number of neighbors, atom charge, and bonded hydrogen atoms to identify its functional group. All input molecules are canonicalized using RDKit prior to running this algorithm, ensuring a deterministic atom ordering. This approach ensures precise detection that strictly aligns with the chemical definitions of functional groups within molecular graphs. The pseudocode is provided in Algorithm 1, and the time complexity of the algorithm is provided in Appendix B.2.

For instance, a carbon atom with a charge of 0 and three neighbors, including a double-bonded oxygen atom, is classified as part of a ketone group (RCOR'), with the oxygen also contributing to the group. Additionally, we address cases where a functional group may be a subset of another. The algorithm first checks for the presence of the larger functional group. If it's not identified, the algorithm then checks for smaller functional groups, ensuring correct identification even in complex structures. Appendix B.1 provides a full list of 101 non-ring-containing functional groups.

The results of the FG annotation algorithm are utilized for two distinct processes:

- **FG-aware tokenization:** The molecular graph, where each node (atom) is assigned to a specific functional group, is converted back into a SMILES string that incorporates functional group information. For example, a ketone-containing group originally depicted in SMILES as *C(=O)* is transformed into an FG-enhanced SMILES string like *C_ketone(=O_ketone)*. This FG-enhanced SMILES embeds additional chemical context directly into the molecular representation while remaining fully compliant with traditional SMILES rules (if the FG information is removed, the FG-enhanced SMILES reverts to its standard SMILES form).

- **FG-aware fragmentation:** Once FGs are identified, the molecule is segmented according to the bonds connecting these groups, as illustrated in Figure 2(b). These bonds serve as edges in a graph, with nodes corresponding to the functional groups. This representation, referred to as the FG graph, captures the molecule's structural backbone.

### 3.2 Atom-Level Representation Learning

We employ a masked language model architecture that takes FG-enhanced SMILES as input, using atom-level tokenization and leveraging masked atom prediction as a self-supervised task to train the model. Specifically, we adopt BERT model (Devlin, 2018) with the following loss:

$$\mathcal{L}_{\mathrm{MLM}} = -\sum_{i \in \mathcal{M}} \log P_\theta(x_i \mid x_{\backslash i}),$$

where $\mathcal{M}$ denotes the set of indices corresponding to the masked tokens, $x_i$ is the original token at position $i$, and $x_{\backslash i}$ represents the input sequence with the token at position $i$ masked. To evaluate the impact of masking, we conduct experiments with varying masking percentages and test the model's performance on six downstream tasks. The results indicate that a masking percentage of 35% yields the highest average performance across tasks. Detailed results and information about the training process are provided in Appendix D.1.

We examine the attention mechanism of the BERT model trained with FG-enhanced SMILES by visualizing the attention scores for a query atom (Figure 3). The attention map reveals that the model pays more attention to atoms that are strongly connected to the query atom than to those that are merely adjacent in the SMILES string. In detail, the query atom at position 23 shows higher attention to the atom at position 0, which is part of the same ring, rather than to the atom at position 26, which is closer in the SMILES string but not directly connected. This demonstrates that the model effectively learns the syntax and semantics of SMILES, capturing the underlying molecular structure rather than merely relying on the linear sequence of SMILES.

Table 1: Evaluation of the FG-enhanced SMILES lexicon size across various databases. "Sequence length" reports the (min; max) number of tokens per molecule. "Vocab size" reports the number of unique tokens included in the lexicon at two minimum frequency thresholds.

| Database | #Molecules | #Atom types | Sequence length (min; max) | Vocab size (freq $\geq 1$) | Vocab size (freq $\geq 5$) |
|---|---|---|---|---|---|
| ZINC15 | 3M | 10 | (5; 63) | 1,151 | 1,089 |
| ChEMBL25 | 1.8M | 35 | (1; 867) | 15,269 | 10,016 |
| Collected dataset | 20M | 46 | (1; 867) | 22,364 | 14,741 |

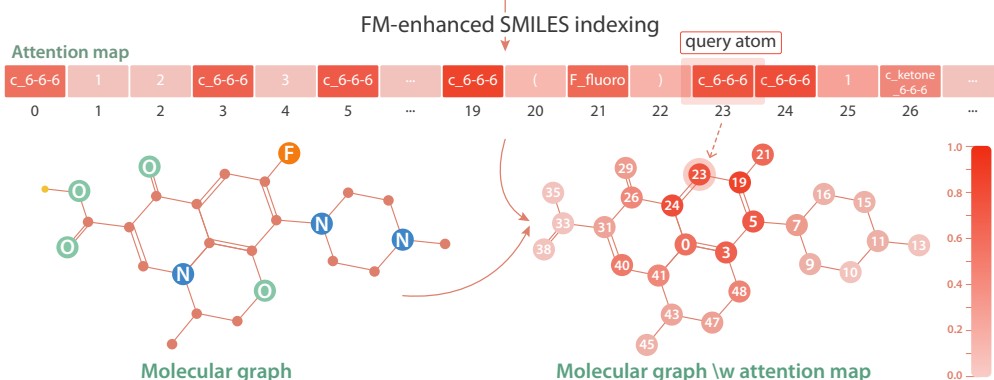

Figure 3: Visualization of the attention map of the BERT model trained with functional group-enhanced SMILES, demonstrating the model's ability to learn and recognize the syntax of SMILES.

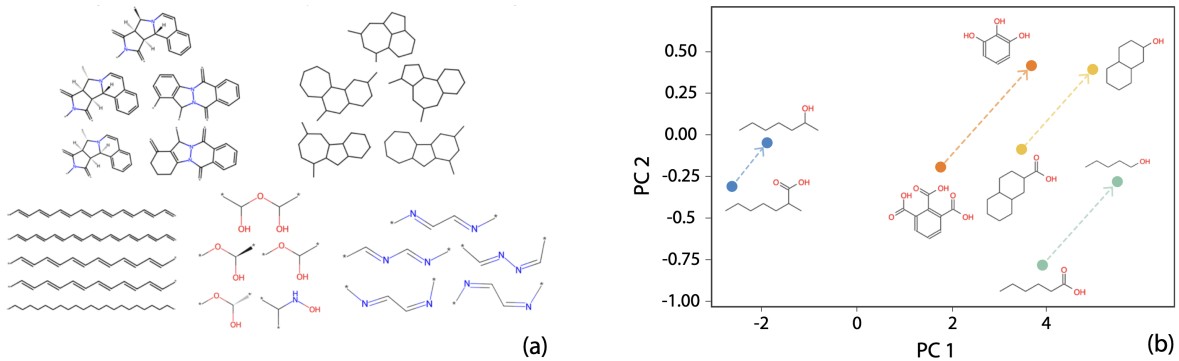

Figure 4: (a) Visualization of functional group knowledge graph embedding space: Clusters of five functional groups with closely related embeddings. (b) Link prediction performance: Substituting one functional group in a molecule with another generates parallel results across different molecules.

### 3.3 Molecular Structure Learning

To learn the structure of molecules, we employ a two-step process. First, we derive **FG embeddings** from the **FG knowledge graph**. Next, we use these embeddings as inputs for a GNN **link prediction model** to predict interactions between FGs. This approach effectively captures the relationships among FGs, both in terms of their structural characteristics and interactions, thereby facilitating the implicit learning of molecular structure.

***FG knowledge graph.*** The FG knowledge graph models each FG with various relations, such as the atoms, bonds, and ring structures it contains, as well as properties like water solubility, lipophilicity (logP), and more. A comprehensive list of FG knowledge graph relations is provided in the Appendix B.1. A snapshot of the FG knowledge graph is shown in Figure 2(c).

The FG knowledge graph embedding is learned by the ComplEx model (Trouillon et al., 2016) to obtain node embeddings. ComplEx is a matrix factorization model specifically designed to learn embeddings from multi-relational data. It is particularly effective at capturing asymmetric and one-to-many relations, making it well-suited for our FG knowledge graph, as several relation types in the FG knowledge graph are inherently asymmetric and one-to-many. For example, `contain_atom` maps one functional group to multiple atoms (one-to-many), and `contain_atom(ketone, C)` does not imply `contain_atom(C, ketone)` (asymmetric). Each element in a triple $(\mathbf{h}, \mathbf{r}, \mathbf{t})$, where $\mathbf{h}$ is the head entity, $\mathbf{r}$ is the relation, and $\mathbf{t}$ is the tail entity, is represented as a complex vector. The score for a given triple is calculated as $f(\mathbf{h}, \mathbf{r}, \mathbf{t}) = \mathrm{Re}\left(\mathbf{h}^T \mathbf{r} \cdot \mathbf{t}\right)$, where the dot product is performed in complex space. Because complex multiplication is not commutative, $f(\mathbf{h}, \mathbf{r}, \mathbf{t}) \neq f(\mathbf{t}, \mathbf{r}, \mathbf{h})$ in general, naturally capturing directional relationships. ComplEx employs a margin-based ranking loss function

Table 2: Performance comparison of FARM and baseline models on MoleculeNet classification and regression tasks. Classification tasks are evaluated by ROC-AUC (%) ($\uparrow$) and regression tasks by RMSE ($\downarrow$) and MAE ($\downarrow$). The first 11 models incorporate functional group (FG) information.

| | Classification | | | | | | | | Regression (RMSE) | | | Regression (MAE) | |
|---|---|---|---|---|---|---|---|---|---|---|---|---|---|
| | Physiology | | | | | Biophysics | | | Physical chemistry | | | Quantum mechanics | |
| Dataset | BBBP | Tox21 | ToxCast | SIDER | ClinTox | BACE | MUV | HIV | ESOL | FreeSolv | Lipo | QM8 | QM9 |
| #tasks | 1 | 12 | 617 | 27 | 2 | 1 | 17 | 1 | 1 | 1 | 1 | 12 | 3 |
| #samples | 2,039 | 7,831 | 8,575 | 1,427 | 1,478 | 1,513 | 93,807 | 41,127 | 1,128 | 642 | 4,200 | 21,786 | 133,885 |
| MICRO | $84.4_{\pm1.1}$ | $77.0_{\pm0.8}$ | $65.2_{\pm0.8}$ | $56.7_{\pm0.9}$ | $77.0_{\pm2.0}$ | $77.2_{\pm2.0}$ | - | $75.1_{\pm1.1}$ | - | - | - | - | - |
| MGSSL | $69.7_{\pm0.9}$ | $76.5_{\pm0.3}$ | $64.1_{\pm0.7}$ | $61.8_{\pm0.8}$ | $80.7_{\pm2.1}$ | $79.1_{\pm0.9}$ | $78.7_{\pm1.5}$ | $78.8_{\pm1.2}$ | - | - | - | - | - |
| MoleOOD | $71.0_{\pm0.8}$ | - | - | $63.4_{\pm0.7}$ | - | $84.3_{\pm1.1}$ | - | $79.4_{\pm0.5}$ | - | - | - | - | - |
| MCM | $90.0_{\pm3.1}$ | $80.2_{\pm1.5}$ | - | $62.7_{\pm2.8}$ | $65.5_{\pm14}$ | $82.0_{\pm5.5}$ | - | - | - | - | - | - | - |
| HimGNN | $92.8_{\pm2.7}$ | $80.7_{\pm1.7}$ | - | $64.2_{\pm2.3}$ | $91.7_{\pm3.0}$ | $85.6_{\pm3.4}$ | - | - | $0.870_{\pm0.154}$ | $1.921_{\pm0.474}$ | $0.632_{\pm0.016}$ | - | - |
| FG-BERT | $70.2_{\pm0.9}$ | $78.4_{\pm0.8}$ | $63.3_{\pm0.8}$ | $64.0_{\pm0.7}$ | $83.2_{\pm1.6}$ | $84.5_{\pm1.5}$ | $75.3_{\pm2.4}$ | $77.4_{\pm1.0}$ | $0.944_{\pm0.025}$ | - | $0.655_{\pm0.009}$ | - | - |
| HiMol (S) | $71.3_{\pm0.6}$ | $76.0_{\pm0.2}$ | - | $62.5_{\pm0.3}$ | $70.6_{\pm2.1}$ | $84.6_{\pm0.2}$ | - | - | - | - | - | - | - |
| HiMol (L) | $73.2_{\pm0.8}$ | $76.2_{\pm0.3}$ | - | $61.3_{\pm0.5}$ | $80.8_{\pm1.4}$ | $84.3_{\pm0.3}$ | - | - | - | - | - | - | - |
| Mole-BERT | $71.9_{\pm1.6}$ | $76.8_{\pm0.5}$ | $62.8_{\pm1.1}$ | $62.8_{\pm1.1}$ | $78.9_{\pm3.0}$ | $80.8_{\pm1.4}$ | $78.6_{\pm1.8}$ | $78.2_{\pm0.8}$ | $1.015_{\pm0.003}$ | - | $0.676_{\pm0.002}$ | - | - |
| MolCLR$_{\text{GCN}}$ | $73.8_{\pm0.2}$ | $74.7_{\pm0.8}$ | - | $\mathbf{66.9}_{\pm1.2}$ | $86.7_{\pm1.0}$ | $78.8_{\pm0.5}$ | $84.0_{\pm1.8}$ | $77.8_{\pm0.5}$ | $1.160_{\pm0.000}$ | $2.390_{\pm0.140}$ | $0.780_{\pm0.010}$ | $0.0181_{\pm0.0002}$ | $0.00445_{\pm0.00002}$ |
| MolCLR$_{\text{GIN}}$ | $73.6_{\pm0.5}$ | $79.8_{\pm0.7}$ | - | $68.0_{\pm1.1}$ | $\mathbf{93.2}_{\pm1.7}$ | $89.0_{\pm0.3}$ | $\mathbf{88.6}_{\pm2.2}$ | $80.6_{\pm1.1}$ | $1.110_{\pm0.010}$ | $2.200_{\pm0.200}$ | $0.650_{\pm0.080}$ | $0.0174_{\pm0.0013}$ | $\mathbf{0.00375}_{\pm0.00000}$ |
| GLAD | $80.4_{\pm1.5}$ | - | - | $64.7_{\pm1.8}$ | $87.3_{\pm1.2}$ | $85.7_{\pm0.9}$ | - | - | - | - | - | - | - |
| N-Gram | $91.2_{\pm3.0}$ | $76.9_{\pm2.7}$ | - | $63.2_{\pm0.5}$ | $85.5_{\pm3.7}$ | $87.6_{\pm3.5}$ | $81.6_{\pm1.9}$ | $83.0_{\pm1.3}$ | $1.100_{\pm0.030}$ | $2.510_{\pm0.190}$ | $0.880_{\pm0.120}$ | $0.0320_{\pm0.003}$ | - |
| Hu et al. | $70.8_{\pm1.5}$ | $78.7_{\pm0.4}$ | $66.5_{\pm0.3}$ | $62.7_{\pm0.8}$ | $72.6_{\pm1.5}$ | $84.5_{\pm0.7}$ | $81.3_{\pm2.1}$ | $79.9_{\pm0.7}$ | $1.100_{\pm0.030}$ | $2.510_{\pm0.191}$ | $0.880_{\pm0.121}$ | $0.0320_{\pm0.003}$ | $0.00964_{\pm0.00031}$ |
| GROVER | $86.8_{\pm2.2}$ | $80.3_{\pm2.0}$ | $65.3_{\pm0.5}$ | $61.2_{\pm2.5}$ | $70.3_{\pm13.7}$ | $82.4_{\pm3.6}$ | $67.3_{\pm3.3}$ | $68.2_{\pm1.1}$ | $1.423_{\pm0.288}$ | $2.947_{\pm0.615}$ | $0.823_{\pm0.010}$ | $0.0182_{\pm0.001}$ | $0.00719_{\pm0.00208}$ |
| GraphMVP | $72.4_{\pm1.6}$ | $75.9_{\pm0.5}$ | $63.1_{\pm0.4}$ | $63.1_{\pm0.4}$ | $79.1_{\pm2.8}$ | $81.2_{\pm0.9}$ | $77.7_{\pm0.6}$ | $77.0_{\pm1.2}$ | - | - | - | - | - |
| GEM | $88.8_{\pm0.4}$ | $78.1_{\pm0.4}$ | $69.2_{\pm0.4}$ | $63.2_{\pm1.5}$ | $90.3_{\pm0.7}$ | $87.9_{\pm1.1}$ | $75.3_{\pm1.5}$ | $81.3_{\pm0.3}$ | $0.813_{\pm0.028}$ | $1.748_{\pm0.114}$ | $0.674_{\pm0.022}$ | $0.0163_{\pm0.001}$ | $0.00562_{\pm0.00007}$ |
| UniMol | $72.9_{\pm0.6}$ | $79.6_{\pm0.5}$ | $69.6_{\pm0.1}$ | $65.9_{\pm1.3}$ | $91.9_{\pm1.8}$ | $85.7_{\pm0.2}$ | $82.1_{\pm1.3}$ | $82.8_{\pm0.3}$ | $0.788_{\pm0.029}$ | $1.480_{\pm0.048}$ | $\mathbf{0.603}_{\pm0.010}$ | $0.0156_{\pm0.001}$ | $0.00467_{\pm0.00004}$ |
| **FARM (Ours)** | $\mathbf{93.3}_{\pm0.2}$ | $\mathbf{81.9}_{\pm1.2}$ | $\mathbf{69.9}_{\pm0.5}$ | $65.9_{\pm0.7}$ | $86.0_{\pm0.8}$ | $\mathbf{89.6}_{\pm0.4}$ | $82.7_{\pm2.1}$ | $\mathbf{83.5}_{\pm0.5}$ | $\mathbf{0.761}_{\pm0.031}$ | $\mathbf{1.097}_{\pm0.033}$ | $0.778_{\pm0.005}$ | $\mathbf{0.0146}_{\pm0.001}$ | $0.00456_{\pm0.00001}$ |

defined as:

$$\mathcal{L}_{\text{Graph}} = \sum_{\substack{(\mathbf{h},\mathbf{r},\mathbf{t}) \\ \in E^+}} \sum_{\substack{(\mathbf{h'},\mathbf{r},\mathbf{t'}) \\ \in E^-}} \max\left(0, \gamma + f(\mathbf{h'},\mathbf{r},\mathbf{t'}) - f(\mathbf{h},\mathbf{r},\mathbf{t})\right) \tag{1}$$

where $E^+$ denotes the set of positive triples, $E^-$ denotes the set of negative triples, and $\gamma$ represents the margin. Optimizing this loss function will maximize the score of positive triples $(\mathbf{h},\mathbf{r},\mathbf{t})$, while minimizing the score of negative triples $(\mathbf{h'},\mathbf{r},\mathbf{t'})$, with a margin $\gamma$ separating the two. This is achieved through margin-based ranking loss, where the function $f(\mathbf{h},\mathbf{r},\mathbf{t})$ evaluates the plausibility of the triples. The goal is to ensure that the score for positive triples is higher than that for negative triples by at least the margin $\gamma$, thus pushing positive triples closer and negative triples farther apart in the embedding space. The detailed implementation of ComplEx is described in Appendix D.2.

By embedding FGs through the knowledge graph, the model can capture both structural and property-based features of each FG, leading to richer FG representations. Figure 4(a) illustrates clusters of FGs in the FG knowledge graph embedding space, showing that similar FGs are closely positioned, suggesting effective structural and property-based grouping.

***Link prediction.*** For link prediction using a graph convolutional network (GCN), the FG graph is used as input. We employ embeddings from the FG knowledge graph as node features for the GCN. During training, node embeddings are computed through graph convolution:

$$\mathbf{h}_i' = \text{ReLU}\left(\mathbf{W} \cdot \frac{1}{|\tilde{\mathcal{N}}(i)|} \sum_{j \in \tilde{\mathcal{N}}(i)} \mathbf{h}_j\right)$$

where $\mathbf{h}_i'$ is the updated embedding for node $i$, computed by averaging the embeddings $\mathbf{h}_j$ of its neighbors $\tilde{\mathcal{N}}(i)$, including itself due to the added self-loop, applying the weight matrix $\mathbf{W}$, and passing through a ReLU activation function.

The score estimates the probability of connections between nodes is computed with a multi-layer perceptron (MLP) and a sigmoid function: $p_{ij} = \sigma(\text{MLP}(\mathbf{h}_i \oplus \mathbf{h}_j))$, where $\oplus$ indicates the concatenation operation.

We optimize the model using binary cross-entropy loss defined as: We then sample positive edges $E^+$ and negative edges $E^-$, and optimize the model using binary cross-entropy loss defined as:

$$\mathcal{L}_{\text{Link}} = -\frac{1}{|E^+|} \sum_{(i,j) \in E^+} \log p_{ij} - \frac{1}{|E^-|} \sum_{(i,j) \in E^-} \log(1 - p_{ij})$$

Figure 4(b) and Figure 8 (Appendix D.3) demonstrates the capability of our molecular structure representation model, showing that, akin to word pair analogy tasks in NLP, replacing one functional group in a molecule with another (in this case, replacing -OH with -COOH) produces parallel results across different molecules. This demonstrates the model's ability to effectively capture and preserve chemical analogies, highlighting its robustness in learning and representing molecular structures.

### 3.4 Molecular Structure Integration via Contrastive Learning

To integrate atom-level representations with molecular structure information, we employ contrastive learning to align FG-enhanced SMILES representations with FG graph embeddings. This method captures both the atom-level and topological aspects of molecular structures, allowing the model to develop a unified representation that integrates chemical context with overall molecular architecture.

In this framework, each molecule is treated as a pair of representations: the FG-enhanced SMILES and its corresponding FG graph. The contrastive learning task encourages the embeddings of these two representations to be as similar as possible for the same molecule, while pushing apart the representations of different molecules. This allows the model to capture both local chemical features (from the FG-enhanced SMILES) and global molecular topology (from the FG graph).

To enhance the learning process and make it more robust, we generate negative examples by augmenting the FG graph. We apply two types of augmentations: (1) node deletion, where one or more functional groups are removed from the graph, and (2) node swapping, where functional groups are randomly exchanged with one another. Figure 2(d) illustrates how these augmentations are applied to generate negative examples from a FG graph. These augmentations create harder negative examples that force the model to better understand the correct structure and connectivity between functional groups, making it more effective at learning meaningful molecular representations.

Specifically, given a positive pair $(\mathbf{h}_{\text{MLM}}, \mathbf{h}_{\text{pos}})$, where $\mathbf{h}_{\text{MLM}}$ is the atom-level representation derived from a pretrained BERT model and $\mathbf{h}_{\text{pos}}$ is the corresponding structure representation from a graph neural network (GNN), and a negative pair $(\mathbf{h}_{\text{MLM}}, \mathbf{h}_{\text{neg}})$, where $\mathbf{h}_{\text{neg}}$ is a augmented FG-graph, the contrastive loss can be written as:

$$\mathcal{L}_{\text{CL}} = \frac{1}{N} \sum_{i=1}^{N} \frac{1}{K} \sum_{j=1}^{K} \max \left( 0, \ \gamma - \cos \left( \mathbf{h}_{\text{MLM}}^{(i)}, \mathbf{h}_{\text{pos}}^{(i)} \right) + \cos \left( \mathbf{h}_{\text{MLM}}^{(i)}, \mathbf{h}_{\text{neg}}^{(j)} \right) \right)$$

where $\gamma$ is the margin parameter and $N$ is the number of training examples (or contrastive pairs). The final objective function for integration is: $\mathcal{L}_{\text{Integration}} = \lambda_{\text{MLM}} \cdot \mathcal{L}_{\text{MLM}} + \lambda_{\text{CL}} \cdot \mathcal{L}_{\text{CL}}$, where $\lambda_{\text{MLM}}$ and $\lambda_{\text{CL}}$ are hyperparameters that control the contribution of each loss to the overall objective. Details on the implementation of contrastive training can be found in the Appendix D.4.

The effect of the contrastive learning objective can be observed in Figure 4(b) and Figure 8. The learned embedding space exhibits meaningful chemical relationships between molecules. The learned embedding space exhibits meaningful chemical relationships between molecules. In particular, similar to the parallel relationships observed in word embeddings in natural language processing, such as *king : queen* being analogous to *man : woman* or *husband : wife*, substituting one functional group with another produces consistent directional shifts in the molecular embedding space across different molecules. This suggests that functional group substitutions correspond to approximately parallel transformations in the learned representation space, indicating that the model captures systematic chemical relationships between functional groups and their structural contexts.

## 4    Pre-training Data Collection and Diversity Assessment

The performance and generalizability of a machine learning model is heavily dependent on the quality and diversity of its training data. Given the vastness of chemical space, it is vital that the training data represents a sufficiently representative subset of this space. Traditional approaches assess the diversity of datasets based on criteria such as the number of chemical elements and the number of atoms per molecule. However, to the best of our knowledge, no previous studies have systematically evaluated the diversity of large chemical datasets like ZINC (Irwin and Shoichet, 2005) or ChEMBL (Gaulton et al., 2011) in terms of functional groups. This gap is significant, as functional groups provide deep insights into the structural and functional complexity of molecules.

In this work, we introduce a novel criterion to assess dataset diversity by using the size of the FG-enhanced SMILES lexicon as a metric. By analyzing the size of this lexicon, we can assess whether the dataset captures a comprehensive range of chemical functionalities, which is crucial for building robust molecular foundation models. Table 1 presents the size of the FG-enhanced SMILES lexicon for the ZINC15 and ChEMBL25 datasets, as well as the dataset we collected for training our foundation model. The table shows that despite its widespread use for training foundation models in small molecule representation, ZINC15 is significantly less diverse compared to ChEMBL25. This is largely due to the fact that ZINC15 is primarily designed to include molecules that adhere to Lipinski's Rule of Five for drug-likeness, excluding more exotic or less common elements that are less relevant to pharmaceutical chemistry. This limited diversity may negatively impact the model's performance on out-of-distribution datasets, which contain a broader range of atom types and functional groups not present in ZINC15. In contrast, ChEMBL25 is far more diverse and thus better suited for training foundation models that can be fine-tuned for various downstream tasks. Based on this insight, we collected a dataset that includes the entire ChEMBL25 database, libraries from chemical drug suppliers, and a subset of ZINC15 to ensure a more comprehensive coverage of chemical space. Detailed information about the collected data can be found in the Appendix A.

Table 3: Performance of FARM on the photostability dataset for quantum mechanics property prediction. $R^2$ scores are reported with standard deviation over 10 runs.

| Class | $R^2$ | Class | $R^2$ | Class | $R^2$ | Class | $R^2$ |
|---|---|---|---|---|---|---|---|
| HOMOm1 (eV) | $0.927 \pm 0.015$ | LUMOp1 (eV) | $0.922 \pm 0.028$ | T3 | $0.900 \pm 0.010$ | S3 | $0.911 \pm 0.008$ |
| HOMO (eV) | $0.895 \pm 0.030$ | DipoleMoment (D) | $0.755 \pm 0.053$ | T4 | $0.921 \pm 0.020$ | S4 | $0.919 \pm 0.008$ |
| LUMO (eV) | $0.928 \pm 0.009$ | LogP | $0.995 \pm 0.006$ | T5 | $0.905 \pm 0.032$ | S5 | $0.924 \pm 0.014$ |
| PrimeExcite (eV) | $0.877 \pm 0.049$ | T1 | $0.929 \pm 0.006$ | S1 | $0.936 \pm 0.016$ | O1 | $0.839 \pm 0.034$ |
| PrimeExcite (osc) | $0.837 \pm 0.024$ | T2 | $0.925 \pm 0.018$ | S2 | $0.948 \pm 0.016$ | TDOS2.0 | $0.868 \pm 0.099$ |
| TDOS2.5 | $0.877 \pm 0.004$ | TDOS3.0 | $0.815 \pm 0.045$ | TDOS3.5 | $0.895 \pm 0.033$ | TDOS4.0 | $0.919 \pm 0.031$ |
| TDOS4.5 | $0.938 \pm 0.018$ | | | | | | |

## 5    Experimental Results

### 5.1    Data, Splits and Evaluation Metrics

**MoleculeNet**  We consider 13 benchmark tasks in the MoleculeNet dataset (Wu et al., 2018). Following previous works, the data is split using a scaffold split into training, validation, and test sets with an 8:1:1 ratio, ensuring fair and consistent evaluation across all models. For all downstream tasks, FARM is used as a frozen feature extractor paired with a GRU prediction head; full finetuning details are provided in Appendix D.5.

We use ROC-AUC for classification tasks, RMSE for physical chemistry tasks (ESOL, FreeSolv, and Lipophilicity), and MAE for quantum mechanics tasks (QM8 and QM9), following previous works. For QM9, we report the average MAE across three targets: HOMO energy, LUMO energy, and HOMO-LUMO gap. For each task, we run three random seeds and report the average and standard deviation.

**D-B-A Oligomer Photostability Dataset**  (Angello et al., 2024) This dataset comprises a chemical library of light-harvesting donor–bridge–acceptor (D–B–A) oligomers, constructed using a modular approach with 22 donor–bridge blocks and 100 acceptor blocks. This design results in a theoretical chemical space of 2,200 unique molecules, which were synthesized and experimentally characterized to study photostability in

solution. We adopt a hard-case scaffold split: 3 donor–bridge blocks and 15 acceptor blocks are held out to create the test set, including all molecules containing any of the held-out donors or acceptors, ensuring that no molecules in the training or validation sets share the same donor or acceptor structures. A similar procedure is used to construct the validation set, and the process is repeated for ten random seeds to ensure robust evaluation across multiple train/validation/test splits.

The tasks include the prediction of frontier molecular orbital energies (HOMO, LUMO, HOMO-1, LUMO+1), dipole moment, logarithm of the partition coefficient (LogP), excitation energies (PrimeExcite), oscillator strengths (PrimeExcite (osc)), triplet and singlet excitation energies (T1–T5, S1–S5), and time-dependent density of states (TDOS 2.0–4.5). We use $R^2$ as the evaluation metric to quantify the accuracy of predicted properties against experimental or high-level computational values. Models are trained on the training set and validated on the held-out validation set, with performance reported on the test set across all ten random scaffold splits to ensure robustness and assess generalization to unseen donor or acceptor blocks.

### 5.2 Baselines

**MoleculeNet**  We consider works that incorporate functional group information to enhance representation learning (Zhang et al., 2020; 2021; Yang et al., 2022; Wang et al., 2023; Han et al., 2023; Li et al., 2023; Zang et al., 2023; Wang et al., 2022b; Xia et al., 2023; Nguyen et al., 2024; Kengkanna and Ohue, 2024), and other approaches that utilize masked atom prediction as a self-supervised training task (Rong et al., 2020; Hu et al., 2019; Liu et al., 2021; Fang et al., 2022; Zhou et al., 2023; Soares et al., 2024).

**D-B-A Oligomer Photostability**  For this dataset, we do not find any prior benchmark studies, as most foundation models for small molecules have focused exclusively on MoleculeNet. We include this dataset to evaluate our model on a diverse set of tasks and to assess its generalizability beyond standard benchmarks.

### 5.3 Results

**MoleculeNet**  Table 2 presents the performance of FARM alongside other baseline models on 13 MoleculeNet tasks. FARM consistently outperforms baseline models across the benchmarks, demonstrating its robustness and versatility. Notably, FARM achieves particularly strong performance on BBBP and BACE, which are often more learnable because these tasks involve well-defined functional group patterns associated with blood–brain barrier penetration and beta-secretase inhibition, respectively—patterns that FG-aware tokenization captures effectively. FARM also excels on solubility-related tasks such as FreeSolv, and ESOL, likely due to its ability to model atom-level chemical context and functional groups that strongly influence solubility. Overall, its strong performance across both classification and regression tasks highlights FARM as a highly effective pretrained model, well-suited for a broad range of downstream applications in molecular property prediction.

**D-B-A Oligomer Photostability**  We evaluate FARM on the D–B–A oligomer photostability dataset across 25 property prediction tasks (results shown in Table 3 and Table 11). The model achieves a lowest $R^2$ of 0.815 for TDOS 3.0 and a highest $R^2$ of 0.995 for LogP, with 16 out of 25 tasks exceeding $R^2$ values of 0.9. These tasks are particularly challenging not only due to the hard-case scaffold split, in which the training set contains no molecules sharing donor or acceptor blocks with the validation and test sets, but also due to the nature of the targets, including difficult-to-predict properties such as dipole moment, excited-state energies (S1–S5 and T1–T5), and total density of states (TDOS). Despite these challenges, FARM demonstrates consistently high performance across all properties. These results highlight FARM's ability to accurately model complex quantum mechanical behaviors by leveraging functional group information and capturing nuanced molecular interactions, underscoring its robustness and generalizability for real-world chemical property prediction tasks.

### 5.4 Ablation Study on FARM Component Analysis

To assess the contribution of each component in our architecture: FG-enhanced SMILES modeling via BERT, FG embeddings from the FG knowledge graph, and molecule backbone modeling using FG graphs with GNN link prediction, we conducted a comprehensive ablation study across several MoleculeNet benchmark tasks (Table 4). The results demonstrate that each component of FARM improves performance over the

baseline, and the combination of all components yields the strongest overall performance across all test tasks. Specifically, incorporating FG-aware tokenization (FG BERT vs. BERT) leads to a substantial improvement across all tasks, highlighting the importance of injecting chemical context into the molecular representation. Further integrating molecular structure via contrastive learning (**FARM** vs. FG BERT) consistently achieves the highest performance across 6 out of 7 downstream tasks, demonstrating the complementary nature of the two representations. Note that random splitting is used consistently across models in this ablation study, while scaffold splitting is applied for benchmarking in Tables 2 to ensure fair comparison with other methods.

Table 4: Performance of various models across 6 MoleculeNet tasks. The data is split using a random split into training, validation, and test sets with an 8:1:1 ratio.

| Model | Description | BBBP | BACE | HIV | Avg | ESOL | FreeSolv | Avg | QM9 |
|---|---|---|---|---|---|---|---|---|---|
| *#tasks* | | *1* | *1* | *1* | | *1* | *1* | | *3* |
| *#samples* | | *2039* | *1513* | *41127* | | *1128* | *642* | | *133885* |
| *Metric* | | | *ROC-AUC* (↑) | | | | *RMSE* (↓) | | *MAE* (↓) |
| **FG_KGE + GAT** | FG knowledge graph embeddings + GAT | $73.23_{\pm1.93}$ | $76.44_{\pm1.27}$ | $71.65_{\pm0.98}$ | 73.77 | $2.35_{\pm0.210}$ | $4.32_{\pm0.29}$ | 3.335 | $0.0139_{\pm0.00014}$ |
| **AttentiveFP** | GNN with masked atom prediction (atom type) | $77.71_{\pm1.30}$ | $77.15_{\pm0.78}$ | $78.81_{\pm0.99}$ | 77.89 | $1.63_{\pm0.042}$ | $2.11_{\pm0.94}$ | 1.87 | $0.0056_{\pm0.00012}$ |
| **FG AttentiveFP** | GNN with masked atom prediction (atom type + FG) | $85.57_{\pm1.32}$ | $87.30_{\pm0.90}$ | $81.21_{\pm0.92}$ | 84.5 | $1.02_{\pm0.034}$ | $1.08_{\pm0.14}$ | 1.05 | $0.0053_{\pm0.00034}$ |
| **BERT** | BERT on canonical SMILES | $82.12_{\pm1.45}$ | $85.12_{\pm0.76}$ | $83.03_{\pm1.12}$ | 83.42 | $1.45_{\pm0.056}$ | $1.89_{\pm0.09}$ | 1.67 | $0.0059_{\pm0.00012}$ |
| **FG BERT** | BERT on FG-enhanced SMILES | $94.36_{\pm0.50}$ | $94.54_{\pm0.40}$ | $81.93_{\pm1.70}$ | 90.27 | $0.608_{\pm0.031}$ | $0.507_{\pm0.03}$ | 0.558 | $0.0041_{\pm0.00017}$ |
| **FARM** | FG BERT + GNN link prediction + contrastive learning | $96.23_{\pm0.7}$ | $96.19_{\pm0.65}$ | $82.13_{\pm1.10}$ | **91.43** | $0.734_{\pm0.039}$ | $0.308_{\pm0.08}$ | **0.521** | **$0.0038_{\pm0.00014}$** |

## 5.5 Ablation Study on Fragmentation Methods

We compare our FG-based fragmentation method with BRICS fragmentation. When treating each BRICS fragment as a functional group, the FG-enhanced SMILES vocabulary expands to 575,612 tokens (min frequency $\leq 5$), compared to just 14,741 tokens with our chemically grounded FG detection method. This difference arises because BRICS generates larger, less reusable fragments, leading to a bloated vocabulary with many rare tokens, which complicates training and increases the risk of out-of-vocabulary issues. As shown in Table 5, our FG-based method consistently outperforms BRICS across all six MoleculeNet tasks under the same setup.

Table 5: Performance comparison of the model using BRICS fragmentation versus our FG-based fragmentation across six MoleculeNet tasks.

| | BBBP | BACE | HIV | | ESOL | FreeSolv | | QM9 |
|---|---|---|---|---|---|---|---|---|
| *#tasks* | *1* | *1* | *1* | **Average** | *1* | *1* | **Average** | *3* |
| *#samples* | *2039* | *1513* | *41127* | | *1128* | *642* | | *133885* |
| *Metric* | | *ROC-AUC* (↑) | | | | *RMSE* (↓) | | *MAE* (↓) |
| **BRICS** | $88.87 \pm 1.5$ | $89.54 \pm 1.09$ | $70.09 \pm 2.24$ | 82.83 | $1.340 \pm 0.235$ | $0.823 \pm 0.14$ | 1.081 | $0.0085 \pm 0.00034$ |
| **OURS** | $96.23 \pm 0.7$ | $96.19 \pm 0.65$ | $82.13 \pm 1.10$ | **91.43** | $0.734 \pm 0.039$ | $0.308 \pm 0.08$ | **0.521** | **$0.0038 \pm 0.00014$** |

## 5.6 Benchmarking on ADMET Tasks

To further demonstrate FARM's generalization beyond MoleculeNet, we evaluate it on a subset of reliable ADMET tasks from the Therapeutics Data Commons (TDC) leaderboard[1], covering Absorption, Distribution, Metabolism, Excretion, and Toxicity properties. We adopt the standardized train/validation/test split from the leaderboard for consistency. As shown in Table 6, FARM achieves state-of-the-art results on 3 out of 7 tasks and demonstrates competitive performance on the remaining tasks. Notably, none of the foundation models included in Table 2 have reported results on these ADMET tasks. We note that many TDC ADMET tasks are limited by small and highly imbalanced datasets, often containing fewer than 100 positive examples, making it difficult to learn generalizable patterns. We therefore evaluate on a subset of more reliable tasks, comparing against all leading models including those whose code is not publicly available. Many leaderboard approaches rely on hand-crafted features and complex pipelines, which are orthogonal to our focus on end-to-end transferable representation learning; combining FARM with such strategies is a promising direction for future work.

---

[1]https://tdcommons.ai/benchmark/admet_group/overview/

Table 6: FARM's performance on ADMET tasks from the TDC leaderboard. Bold indicates state-of-the-art.

| Dataset | Task | Unit | Metric | Type | SOTA | FARM | Rank |
|---------|------|------|--------|------|------|------|------|
| Bioav | Absorption | % | AUROC | Binary | 0.753 | 0.709 | 5 |
| **AqSol** | | log mol/L | MAE | Regression | 0.761 | **0.739** | **1** |
| **PPBR** | Distribution | % | MAE | Regression | 7.526 | **7.376** | **1** |
| VDss | | L/kg | Spearman | Regression | 0.724 | 0.652 | 4 |
| CYP2C9 Inhibition | Metabolism | % | AUPRC | Binary | 0.859 | 0.798 | 4 |
| CYP3A4 Inhibition | | % | AUPRC | Binary | 0.916 | 0.877 | 5 |
| **Ames** | Excretion | % | AUROC | Binary | 0.871 | **0.875** | **1** |

## 6  Conclusion, Limitations, and Future Work

In summary, FARM demonstrates robust performance across various MoleculeNet classification tasks, outperforming or matching baseline models. The integration of functional group information and the alignment of FG-enhanced SMILES representations with FG graph embeddings through contrastive learning significantly enhance its effectiveness. While FARM shows strong performance, there is a key limitation that should be addressed in future work. The current model does not incorporate a full 3D molecular representation, which is critical for capturing stereochemistry and spatial configurations that influence molecular properties. Integrating 3D structural information (Yan et al., 2024) could further improve the model's ability to make accurate predictions.

Looking ahead, our ultimate goal is to develop a pre-trained atom embedding that parallels the capabilities of pre-trained word embeddings in NLP. This would enable a richer and more nuanced understanding of molecular properties and behaviors at the atomic level. Similarly, we aim to achieve molecule-level representations that are as expressive and versatile as sentence-level embeddings in NLP, capturing both local and global molecular features. By bridging the gap between atom-wise embeddings and holistic molecule representations, FARM paves the way for more accurate, generalizable molecular predictions across a variety of tasks.

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

# A    Molecular Datasets

## A.1    Training data

We collected a diverse dataset to train our **FARM** model from various sources, including ChEMBL25, ZINC15, and several chemical suppliers. The number of compounds in each dataset is reported as follows:

Table 7: List of compound suppliers and number of compounds

| Supplier | Number of Compounds | Source |
|---|---|---|
| Targetmol | 22,555 | https://www.targetmol.com/ |
| Chemdiv | 1,741,620 | https://www.chemdiv.com/ |
| Enamine | 862,698 | https://enamine.net/ |
| Life Chemical | 347,657 | https://lifechemicals.com/ |
| Chembridge | 1,405,499 | https://chembridge.com/ |
| Vitas-M | 1,430,135 | https://vitasmlab.biz/ |
| InterBioScreen | 560,564 | https://www.ibscreen.com/ |
| Maybridge | 97,367 | https://chembridge.com/ |
| Asinex | 601,936 | https://www.asinex.com/ |
| Eximed | 61,281 | https://eximedlab.com/ |
| Princeton BioMolecular | 1,647,078 | https://princetonbio.com/ |
| Otava | 9,203,151 | https://www.otava.com/ |
| Alinda Chemical | 733,152 | https://www.alinda.ru/synthes_en.html |
| ChEMBL 25 | 1,785,415 | https://www.ebi.ac.uk/chembl/ |
| ZINC15 | 4,000,000 | https://zinc15.docking.org/ |
| **Total** | 20,000,000 | |

To ensure data quality, we applied standard preprocessing steps: molecules that failed RDKit processing (i.e., invalid SMILES) were removed, duplicate molecules were identified and removed by converting all SMILES to their canonical form using RDKit, and any overlap with the MoleculeNet benchmark datasets used for downstream evaluation was excluded to prevent data leakage.

## A.2    Lexicon Growth for FG-enhanced SMILES Dataset

As part of our investigation into the relationship between corpus size and vocabulary size, we conducted experiments to observe how vocabulary growth evolves as the dataset increases. Our results show that while the vocabulary size increases with the corpus size, it eventually stabilizes. Specifically, when expanding the corpus from 20 million tokens to 200 million tokens, the vocabulary grew from 14,741 to 42,364 tokens, and with a 1 billion-token corpus, the vocabulary reached a plateau of 42,391 tokens. This indicates that the lexicon tends to converge around 42.3K tokens as the corpus size increases, suggesting that further growth in corpus size may not result in substantial increases in vocabulary.

Even with the smaller vocabulary of 14,741 tokens derived from the 20 million-token dataset, the model was able to achieve excellent coverage of common tokens. In fact, the 14,741 tokens from the 20M dataset accounted for 99.997% of all tokens in the 200M dataset, with 2,932,883,078 out of 2,932,984,142 tokens already represented. This demonstrates that a relatively smaller vocabulary can effectively capture the majority of the data, even with larger corpora. Moreover, the ratio of the UNKNOWN token in downstream datasets remains consistently below 0.1

## A.3    Downstream Tasks Data

In Table 8, we provide an overview of the datasets used for evaluating the performance of our model on various downstream tasks. Each dataset is denoted by its name, followed by the number of tasks it encompasses, the total number of samples available in each dataset, and a brief description. These datasets cover a range of chemical and biological properties, enabling comprehensive evaluation of the model's performance across different tasks in molecular representation learning.

Table 8: Overview of downstream tasks, corresponding sample sizes, and dataset descriptions.

| Dataset | # Tasks | # Samples | Description |
|---|---|---|---|
| BBBP | 1 | 2,039 | Benchmark for Blood-Brain Barrier permeability prediction, assessing whether compounds can cross the blood-brain barrier. |
| Tox21 | 12 | 7,831 | Toxicology data containing multiple assays for evaluating the toxicity of compounds across various endpoints. |
| SIDER | 27 | 1,427 | Side Effect Resource dataset that includes drug side effects associated with FDA-approved drugs, focusing on adverse drug reactions. |
| ClinTox | 2 | 1,478 | Clinical Toxicology dataset designed to predict the toxicity of drug-like compounds based on clinical data. |
| BACE | 1 | 1,513 | Data for predicting activity against the beta-secretase enzyme, relevant for Alzheimer's disease drug discovery. |
| MUV | 17 | 93,807 | Multiple Unrelated Variables dataset aimed at assessing the ability to predict various molecular properties and activities. |
| HIV | 1 | 41,127 | Dataset focused on predicting the activity of compounds against the HIV virus, crucial for antiviral drug development. |
| ESOL | 1 | 1,128 | Dataset used for estimating the solubility of organic compounds in water, useful for understanding compound behavior in biological systems. |
| FreeSolv | 1 | 642 | Dataset containing free energy of solvation values for small organic molecules in water, aiding in solvation energy predictions. |
| Lipophilicity | 1 | 4,200 | Data focused on predicting the octanol-water partition coefficient, a key measure of a compound's lipophilicity. |
| QM8 | 12 | 21,786 | Quantum Mechanics dataset that provides a range of molecular properties computed using quantum mechanical methods for small organic molecules. |
| QM9 | 3 | 133,885 | Quantum Mechanics dataset providing molecular properties for a large set of small organic compounds. |

# B   FG-aware Tokenization and Fragmentation

## B.1   The List of Functional Groups

The exhaustive list of 101 functional groups that can be detected by the functional group detection algorithm includes: Tertiary carbon, Quaternary carbon, Alkene carbon, Cyanate, Isocyanate, Hydroxyl, Ether, Hydroperoxy, Peroxy, Haloformyl, Aldehyde, Ketone, Carboxylate, Carboxyl, Ester, Hemiacetal, Acetal, Hemiketal, Ketal, Orthoester, Carbonate ester, Orthocarbonate ester, Amidine, Carbamate, Isothiocyanate, Thioketone, Thial, Carbothioic S-acid, Carbothioic O-acid, Thiolester, Thionoester, Carbodithioic acid, Carbodithio, Trifluoromethyl, Difluorochloromethyl, Bromodifluoromethyl, Trichloromethyl, Bromodichloromethyl, Tribromomethyl, Dibromofluoromethyl, Triiodomethyl, Difluoromethyl, Fluorochloromethyl, Dichloromethyl, Chlorobromomethyl, Chloroiodomethyl, Dibromomethyl, Bromoiodomethyl, Diiodomethyl, Alkyl, Alkene, Alkyne, Carboxylic anhydride, Primary amine, Secondary amine, Amide, Imide, Tertiary amine, 4-ammonium ion, Hydrazone, Primary ketimine, Primary aldimine, Secondary ketimine, Secondary aldimine, Nitrile, Azide, Azo, Nitrate, Isonitrile, Nitrosooxy, Nitro, Nitroso, Aldoxime, Ketoxime, Sulfhydryl, Sulfide, Disulfide, Sulfinyl, Sulfonyl, Sulfur dioxide, Sulfuric acid, Sulfino, Sulfonic acid, Sulfonate ester, Thiocyanate, Phosphino, Phosphono, Phosphate, Phosphodiester, Phosphoryl, Borono, Boronate, Borino, Borinate, Silyl ether, Dichlorosilane, Trimethylsilyl, Fluoro, Chloro, Bromo, Iod.

## B.2   Time Complexity of the FG Annotation Algorithm

In this work, small molecules refer to compounds with molecular weight below 500 Daltons, following the standard definition in drug discovery and cheminformatics. The average number of atoms in our dataset is 25.77. The FG annotation algorithm traverses each molecule twice, once for each of two FG sets: (1) functional groups that contain other functional groups (e.g., COOH containing CO and OH), and (2) the remaining functional groups. Separating into exactly two sets is sufficient because no functional group contains a smaller one that itself contains an even smaller one, so there is no nesting beyond one level. At each atom, the FG matching step does not require a general subgraph isomorphism check. Instead, each FG pattern is defined by a small, fixed set of local constraints on an atom and its immediate neighborhood, specifically atom type, bond types to neighboring atoms, number of neighbors, atom charge, and bonded hydrogen count. This makes each check a bounded local pattern match of cost $O(k)$, where $k$ is the size of the local neighborhood, bounded by the largest FG definition size, rather than a search over the entire molecular graph. The overall complexity is therefore $O(n \times 2 \times k) = O(n \times k)$, which is linear in the number of atoms

$n$. Given the small average molecule size in our dataset, the algorithm is extremely efficient in practice and does not pose any computational bottleneck even for larger molecules.

### B.3 Naming functional groups with Rings in Fused Ring Systems

Fused ring systems are a diverse and prevalent class of functional groups, accounting for 99.37% of the total functional groups in our dataset (147,564 out of 148,507 FGs). Despite their importance, many of these systems lack standardized nomenclature. To address this, we propose a systematic approach to naming these ring systems based on their ring sizes and core structures.

Each ring in a fused ring system is named according to its size. For instance, a six-membered aromatic ring like benzene is named ring_6. This straightforward approach provides a clear identifier for individual rings within a system. For systems composed of multiple fused rings, we use the following naming convention:

- **Identification:** Determine the smallest atom index for each ring within the system.
- **Sorting:** Arrange the rings by increasing atom indices.
- **Construction:** Combine the ring sizes in ascending order. For example, a fused system with a six-membered ring and a five-membered ring would be named ring_5_6.

This systematic naming helps in identifying and categorizing complex fused ring systems by focusing on their core structure. The core structure is defined as the central framework of interconnected rings that forms the fundamental backbone of the molecule. The core structure of a ring system is important because it influences the molecule's reactivity, stability, and biological activity. In SMILES notation, which uses lowercase characters to indicate atoms within aromatic rings, we can enhance the representation by combining the atom symbol (uppercase or lowercase) with the core structure, thereby providing a comprehensive depiction of the ring system. Figure 5a illustrates an example of naming a fused ring system based on the rules described above, and Figure 5b shows how FG-aware tokenization is applied.

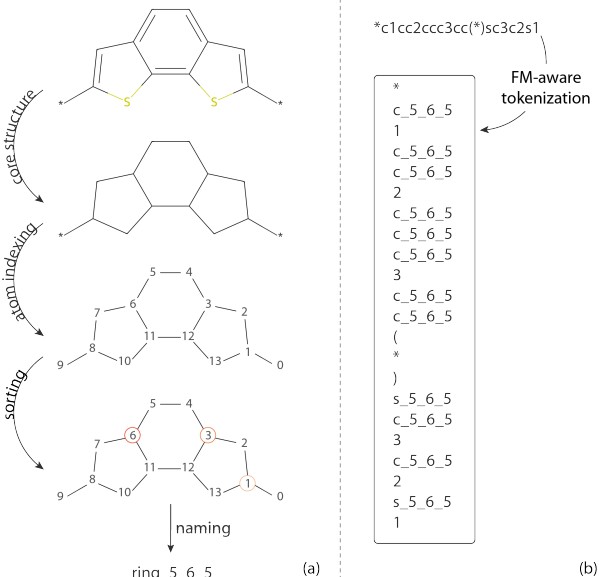

Figure 5: (a) Example of naming a fused ring system in 4 steps: generate the core structure of the functional group, index atoms using RDKit, select the smallest-index atom in each ring and sort, and name the fused ring system based on ring size. (b) Example of FG-aware tokenization.

After completing the naming process, we derive a new FG-enhanced SMILES representation for the molecules. We then analyze our collected dataset, which comprises 20 million samples of FG-enhanced SMILES, to evaluate the results. This dataset includes representations of 46 different elements. Notably, 11 elements are represented by only a single form, indicating their rare occurrence within the dataset (excluding hydrogen). These elements are: H, Ti, V, Cr, Rb, Mo, Rh, Sb, Ba, Pb, and Bi. In contrast, the remaining 35 elements feature at least two representations, each

corresponding to distinct FGs. The distribution of these elements is visualized in Figure 6, highlighting the diversity of representations in our dataset. The most prevalent element in our dataset is Carbon, with 9,112 FGs containing it. Nitrogen follows as the second most prevalent element, represented in 2,549 FGs, while Oxygen and Sulfur appear in 2,156 and 571 FGs, respectively.

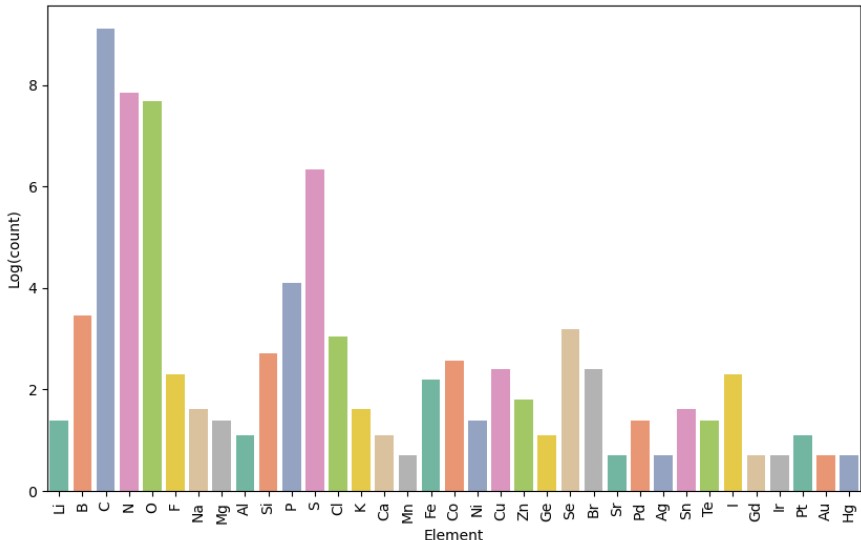

Figure 6: Number of functional groups associated with different chemical elements in the FG-enhanced SMILES dataset. The y-axis represents the natural logarithm (log, base $e$) of the count.

## C  FG Knowledge Graph

The FG knowledge graph is designed to capture both the structural and property-related information of FGs. The list of relations includes:

Table 9: Key relations defined in the FG knowledge graph. (Note: Continuous values, such as LogP and water solubility, are discretized by rounding to the nearest integer.)

| Relation | Description |
| --- | --- |
| contain_atom | Identifies atoms present in the FG (e.g., C, H, O, N). |
| contain_bond | Specifies types of bonds in the FG (e.g., single, double, triple, aromatic). |
| functional_group | Recognizes functional groups in the FG (e.g., hydroxyl, carboxyl, amine). |
| contain_ring_[n] | Indicates the presence of a non-aromatic ring of size n in the FG. |
| contain_aromatic_ring_[n] | Indicates the presence of an aromatic ring of size n in the FG. |
| num_substitutes | Specifies the number of substituents (e.g., alkyl or aryl groups) in the FG. |
| is_hydrogen_bond_donor | Identifies whether the FG contains a functional group capable of donating hydrogen bonds. |
| is_hydrogen_bond_acceptor | Identifies whether the FG contains a functional group capable of accepting hydrogen bonds. |
| logp | Measures the lipophilicity of the FG using the logP value (calculated via RDKit). In the collected dataset, values range from -35 to 31. |
| water_solubility | Predicts the solubility of the FG in water, based on logP, molecular weight, and TPSA. In the collected dataset, values range from -5 to 8. |
| core_smiles | The SMILES representation of the core structure of the FG. |

- **List of functional groups that act as hydrogen bond donors:** Hydroxyl, Hydroperoxy, Primary amine, Secondary amine, Hydrazone, Primary ketimine, Secondary ketimine, Primary aldimine, Amide, Sulfhydryl, Sulfonic acid, Thiolester, Hemiacetal, Hemiketal, Carboxyl, Aldoxime, Ketoxim.

- **List of functional groups that act as hydrogen bond acceptors:** Ether, Peroxy, Haloformyl, Ketone, Aldehyde, Carboxylate, Carboxyl, Ester, Ketal, Carbonate ester, Carboxylic anhydride, Primary amine, Secondary amine, Tertiary amine, 4-Ammonium ion, Hydrazone, Primary ketimine, Secondary ketimine, Primary aldimine, Amide, Sulfhydryl, Sulfonic acid, Thiolester, Aldoxime, Ketoxi.

# D   Implementation Details

## D.1   Training Masked Language Model for SMILES Representation

We trained the BERT model using Hugging Face (Wolf et al., 2020) on the masked molecule prediction task with both conventional SMILES and FG-enhanced SMILES from our collected dataset. To assess the impact of different masking percentages, we trained BERT models with masking percentages of 0.15, 0.25, 0.35, 0.45, and 0.55. The models were then evaluated on seven MoleculeNet tasks, including three classification tasks and four regression tasks, to determine the optimal masking percentage. The results, presented in Table 10, indicate that a masking percentage of 0.35 yields the best performance across the considered downstream tasks.

Table 10: Performance of BERT models with varying masking percentages across six MoleculeNet tasks. The data is split using a random split into training, validation, and test sets with an 8:1:1 ratio.

| | BBBP | BACE | HIV | Average | ESOL | FreeSolv | Average | QM9 |
|---|---|---|---|---|---|---|---|---|
| *#tasks* | *1* | *1* | *1* | | *1* | *1* | | *3* |
| *#samples* | *2039* | *1513* | *41127* | | *1128* | *642* | | *133885* |
| *Metric* | *ROC-AUC (↑)* | | | | *RMSE (↓)* | | | *MAE (↓)* |
| **0.25** | $93.01 \pm 0.9$ | $94.31 \pm 1.08$ | $80.17 \pm 1.5$ | 89.16 | $0.688 \pm 0.033$ | $0.622 \pm 0.007$ | 0.655 | $0.0091 \pm 0.00001$ |
| **0.25** | $93.59 \pm 1.7$ | $93.94 \pm 1.4$ | $81.03 \pm 1.9$ | 89.52 | $0.543 \pm 0.030$ | $0.714 \pm 0.010$ | 0.629 | $\mathbf{0.0032 \pm 0.00001}$ |
| **0.35** | $94.36 \pm 0.5$ | $94.54 \pm 0.4$ | $81.93 \pm 1.7$ | **90.27** | $0.608 \pm 0.031$ | $0.507 \pm 0.030$ | **0.558** | $0.0041 \pm 0.00001$ |
| **0.45** | $93.48 \pm 1.3$ | $94.36 \pm 0.90$ | $80.12 \pm 1.7$ | 89.32 | $0.795 \pm 0.028$ | $0.493 \pm 0.008$ | 0.644 | $0.0048 \pm 0.00001$ |
| **0.55** | $92.85 \pm 1.1$ | $88.68 \pm 1.0$ | $79.89 \pm 0.90$ | 87.14 | $0.734 \pm 0.030$ | $0.599 \pm 0.005$ | 0.667 | $0.0097 \pm 0.00001$ |

Additional details of the training setup include training the BERT model on 20 million SMILES for 15 epochs using two NVIDIA Tesla V100 GPUs. The learning rate was set to $1e-5$ , with a batch size of 128, and model checkpoints were saved after every 10,000 batches. This setup was also applied to the baseline model, which used conventional SMILES for comparison.

Figure 7 illustrates the convergence behavior of the models trained on different representations of molecular data. The model utilizing FG-enhanced SMILES exhibits a slower convergence rate, attributed to the increased complexity of its vocabulary, reflecting its closer resemblance to natural language. The SMILES model converges by step 200 (after processing 25,600 SMILES), while the FG-enhanced SMILES model achieves convergence by step 300 (after processing 38,400 SMILES). Notably, despite the larger prediction vocabulary (14,714 vs. 93), the FG-enhanced model ultimately reaches a lower loss, suggesting its enhanced capacity to capture intricate molecular representations and improve generalization in complex tasks. This indicates the model's ability to leverage functional group information effectively, potentially leading to better performance in downstream applications.

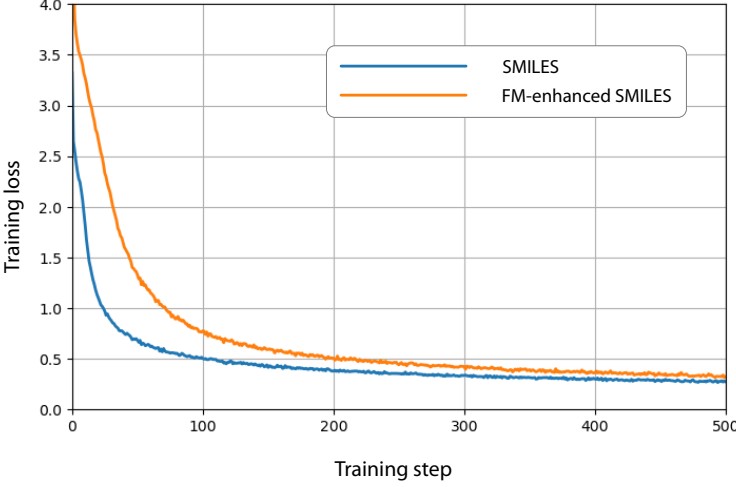

Figure 7: Loss curves for the masked language model (MLM) during training on two datasets: standard SMILES and functional group-enhanced SMILES.

$$\mathcal{L}_{\text{Graph}} = \sum_{(h,r,t)\in E^+} \sum_{(\mathbf{h}',\mathbf{r},\mathbf{t}')\in E^-} \max\left(0, \gamma + f(\mathbf{h}',\mathbf{r},\mathbf{t}') - f(\mathbf{h},\mathbf{r},\mathbf{t})\right) \tag{4}$$

## D.2 Training FG Knowledge Graph Embedding Model for Molecular Structure Representation

Once the FG knowledge graph is constructed as detailed in Section C, we utilize the ComplEx model to learn embeddings for the functional groups. The knowledge graph comprises 148,507 unique nodes: 147,564 corresponding to ring systems and 943 representing non-ring functional groups. Training is conducted with a batch size of 64, a learning rate of $1 \times 10^{-3}$, over 50 epochs, with model checkpoints saved at the end of each epoch.

**ComplEx Model Representation**

In the ComplEx model (Trouillon et al., 2016), each element in a triple $(\mathbf{h}, \mathbf{r}, \mathbf{t})$ — where $\mathbf{h}$ is the head entity, $\mathbf{r}$ is the relation, and $\mathbf{t}$ is the tail entity — is represented as a complex vector:

$$\mathbf{h}, \mathbf{r}, \mathbf{t} \in \mathbb{C}^d \tag{2}$$

*Scoring Function*

The score for a given triple $(\mathbf{h}, \mathbf{r}, \mathbf{t})$ is calculated as:

$$f(\mathbf{h}, \mathbf{r}, \mathbf{t}) = \text{Re}\left(\mathbf{h}^T \mathbf{r} \cdot \mathbf{t}\right) \tag{3}$$

where $\mathbf{r}$ is a complex-valued vector, and the dot product is performed in the complex space.

*Loss Function*

ComplEx employs a margin-based ranking loss function defined as:

where $E^+$ denotes the set of positive triples, $E^-$ denotes the set of negative triples, and $\gamma$ represents the margin.

To assess the quality of the learned embeddings, we randomly sample clusters of five closely related embedding vectors and analyze their arrangement in the embedding space. The results of this evaluation are presented in Figure 4a.

## D.3 Link Prediction Model Using GNNs

For link prediction using the GCN model, we start by segmenting molecules into functional groups via FG-aware molecular segmentation, where each group is connected by single bonds. We then use embeddings from the FG knowledge graph embedding model as node features for the GCN. The training process involves computing node embeddings through graph convolution (Equation 5), followed by scoring potential edges with a multi-layer perceptron (MLP) (Equation 6). This score is used to calculate the probability between two nodes (Equation 7). Positive and negative edges are sampled, and the model is optimized to maximize scores for positive edges while minimizing scores for negative edges using the loss function in Equation 8. This approach effectively trains the model to distinguish between likely and unlikely connections between functional groups.

$$\mathbf{h}_i' = \text{ReLU}\left(\mathbf{W} \cdot \frac{1}{|\mathcal{N}(i)|} \sum_{j\in\mathcal{N}(i)} \mathbf{h}_j\right) \tag{5}$$

where $\mathbf{h}_i'$ is the updated embedding for node $i$. It is computed by averaging the embeddings $\mathbf{h}_j$ of neighboring nodes $\mathcal{N}(i)$, applying the weight matrix $\mathbf{W}$, and then passing through the ReLU activation function.

$$s_{ij} = \text{MLP}(\mathbf{h}_i \oplus \mathbf{h}_j) \tag{6}$$

where $s_{ij}$ denotes the score assigned to the potential edge between nodes $i$ and $j$. The score is computed using a multi-layer perceptron (MLP), which takes as input the concatenated node embeddings of $i$ and $j$, denoted as $\mathbf{h}_i \oplus \mathbf{h}_j$. Here, $\mathbf{h}_i$ and $\mathbf{h}_j$ represent the node embeddings for nodes $i$ and $j$, respectively. The operator $\oplus$ indicates the concatenation of these embeddings. The MLP processes this concatenated vector to produce a score that reflects the likelihood of an edge existing between $i$ and $j$.

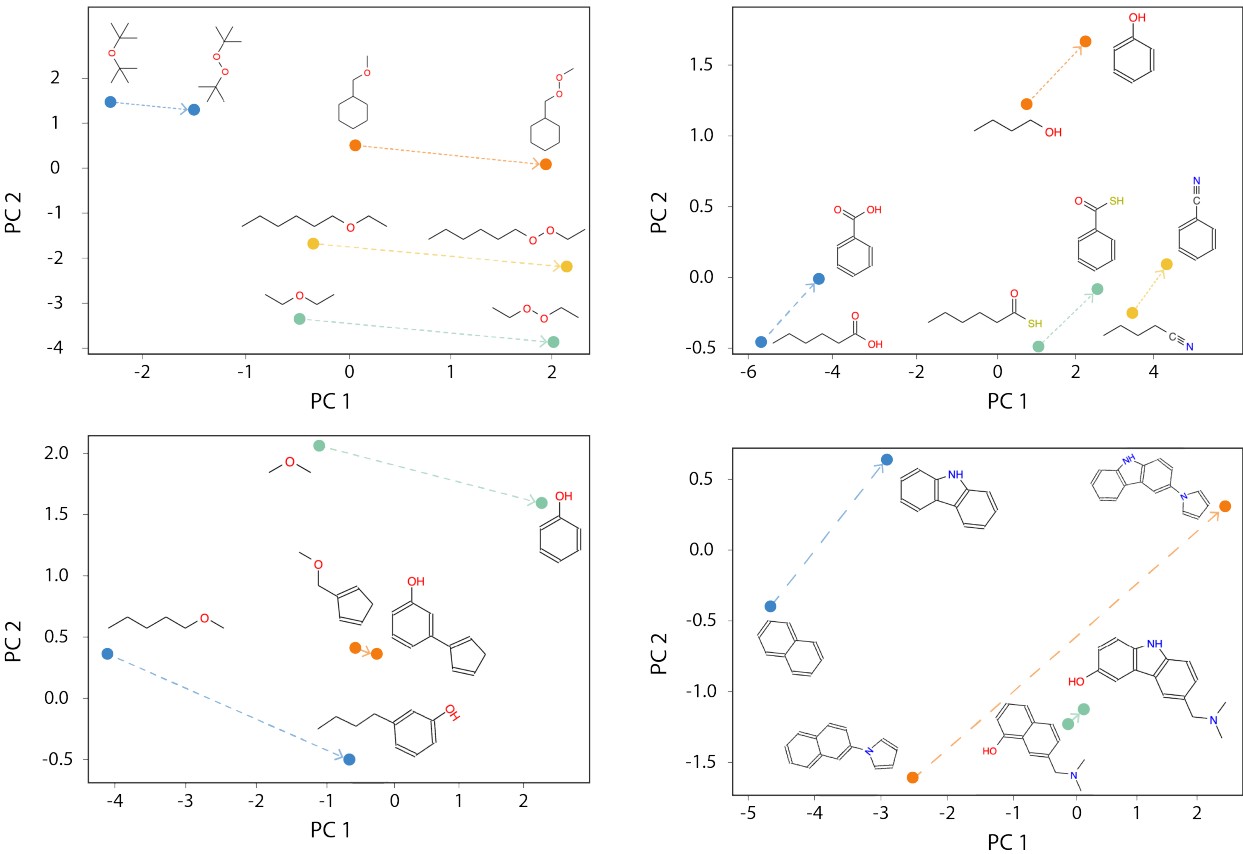

Figure 8: Link prediction model performance: Similar to word embedding analogies in NLP, replacing one functional group in a molecule with another produces parallel results across different molecules, demonstrating the model's ability to capture chemical relationships effectively.

$$p_{ij} = \sigma(s_{ij}) \tag{7}$$

where $\sigma$ is the sigmoid function.

$$\mathcal{L}_{\text{Link}} = -\frac{1}{|E^+|} \sum_{(i,j) \in E^+} \log p_{ij} - \frac{1}{|E^-|} \sum_{(i,j) \in E^-} \log(1 - p_{ij}) \tag{8}$$

where $\mathcal{L}$ is the loss function for link prediction. It computes the average log-likelihood of positive edges $E^+$ and negative edges $E^-$, where $p_{ij}$ is the predicted probability of an edge between nodes $i$ and $j$. The loss penalizes the model for incorrect predictions, encouraging high probabilities for true edges and low probabilities for false edges.

The GCN model for link prediction is trained as follows: For each molecule, represented as a FG graph, we generate all possible combinations of nodes, encompassing both positive pairs (nodes that are linked) and negative pairs (nodes that are not linked). In cases where the graph contains more than three nodes (FGs), we select 60% of all possible combinations along with all positive pairs to form the training data for each graph. The model is subsequently trained for three epochs on a comprehensive dataset consisting of 20 million data points. Figure 8 shows the performance of the link prediction model. Similar to word embedding analogies in NLP, replacing one FG in a molecule with another produces parallel results across different molecules, demonstrating the model's ability to capture chemical relationships effectively.

## D.4 Contrastive Learning: Align SMILES and Structure Representation

In this work, we propose a contrastive learning strategy to align SMILES-based representations of molecules with their corresponding graph-based molecular structures. The goal of this approach is to capture both the sequential

information from SMILES and the structural relationships encoded in graph representations, thus allowing the model to learn a more comprehensive molecular representation that bridges these two modalities.

To measure the similarity between representations derived from the FG-enhanced SMILES and FG graph, we utilize cosine similarity, which is defined as: The cosine similarity between two vectors $\mathbf{u}$ and $\mathbf{v}$ is defined as:

$$\text{cosine\_similarity}(\mathbf{u}, \mathbf{v}) = \frac{\mathbf{u} \cdot \mathbf{v}}{\|\mathbf{u}\|\|\mathbf{v}\|}$$

Here, $\mathbf{u}$ and $\mathbf{v}$ represent the embeddings from two different modalities, such as the SMILES-based BERT output and the GNN output for the molecular graph. This similarity score helps ensure that embeddings of positive (i.e., matched) SMILES and graph representations are closer in the latent space.

To align these two types of representations, we use contrastive loss, a popular technique in self-supervised learning that enforces representations from the same sample (positive pair) to be more similar than those from different samples (negative pair). Given a positive pair $(\mathbf{h}_{\text{MLM}}, \mathbf{h}_{\text{pos}})$, where $\mathbf{h}_{\text{MLM}}$ is the SMILES representation derived from a pretrained BERT model and $\mathbf{h}_{\text{pos}}$ is the corresponding representation from a graph neural network (GNN), and a negative pair $(\mathbf{h}_{\text{MLM}}, \mathbf{h}_{\text{neg}})$, where $\mathbf{h}_{\text{neg}}$ is a augmented FG-graph, the contrastive loss can be written as:

$$\mathcal{L}_{\text{CL}} = \frac{1}{N} \sum_{i=1}^{N} \max\left(0, \gamma - \cos(\mathbf{h}_{\text{MLM}}, \mathbf{h}_{\text{pos}}) + \cos(\mathbf{h}_{\text{MLM}}, \mathbf{h}_{\text{neg}})\right)$$

Where:

- $\gamma$ is the margin parameter, ensuring that the positive similarity is significantly larger than the negative similarity.
- $N$ is the number of training examples (or contrastive pairs)

The objective function is

$$\mathcal{L} = \lambda_{\text{MLM}} \cdot \mathcal{L}_{\text{MLM}} + \lambda_{\text{CL}} \cdot \mathcal{L}_{\text{CL}}$$

where $\mathcal{L}_{\text{MLM}}$ represents the masked language modeling loss, which encourages the model to predict masked tokens in the input sequence effectively, and $\mathcal{L}_{\text{CL}}$ denotes the contrastive loss, which aligns the SMILES and structural representations. The coefficients $\lambda_{\text{MLM}}$ and $\lambda_{\text{CL}}$ are hyperparameters that control the contribution of each loss to the overall objective. By tuning these coefficients, we can balance the learning process between the two tasks, allowing the model to learn rich and meaningful representations from both the sequential and structural aspects of the molecular data.

This combined loss function enables the model to leverage the strengths of both masked language modeling and contrastive learning, fostering a more comprehensive understanding of molecular representations that can enhance performance in downstream tasks such as property prediction, molecular generation, and structure-based drug discovery.

In our contrastive learning model, we set the margin $\gamma = 0.5$ and the weights $\lambda_{MLM} = 1.0$ and $\lambda_{CL} = 0.5$. We train the contrastive BERT model using a batch size of 126 for a total of 5 epochs. This training configuration mirrors the setup used for learning atom representations with the BERT model, as described in Section D.1.

The overall model architecture consists of three main modules that require training: a BERT model, a FG knowledge graph, and GNN. The FG knowledge graph is represented using a ComplEx model and trained first to generate FG embeddings. These embeddings are then used to train the GNN on a link prediction task. Once the GNN training converges, molecular structure embeddings are extracted and a small projection layer is trained to align them with BERT's representations via contrastive learning. This alignment is optimized jointly with the masked language modeling loss and the contrastive loss, updating both the BERT model and the projection layer.

## D.5  Downstream task finetuning

MoleculeNet tasks are treated as downstream tasks for our **FARM** model. We freeze all layers of FARM and pair it with a GRU head for both classification and regression tasks. For classification, we use cross-entropy as the loss function, while for regression, we employ mean squared error. The Adam optimizer is applied with a learning rate of

$1e-4$ and a cosine annealing learning rate schedule with a period of 20 epochs. The training process spans 100 epochs with a batch size of 16, using an 80-10-10 train-validation-test split with scaffold splitting. To address imbalanced datasets, we implement a weighted loss function, assigning a weight of 5 to classes with fewer samples. For each task, we conduct three runs with different train-validation-test splits and report the average and standard deviation of the results.

## E  Ablation Study

### E.1  FARM Component Analysis

To assess the effectiveness of each component in our architecture, we conducted a comprehensive ablation study across several MoleculeNet benchmark tasks. The first model, FG_KGE + GAT, utilizes FG knowledge graph embeddings as input for a Graph Attention Network (Veličković et al., 2017) (GAT) to predict molecular properties. Although its performance on these tasks is not the strongest, the model still demonstrates its capacity to learn underlying chemical rules (syntax and semantics) from the data to a certain degree.

The second model, AttentiveFP (Xiong et al., 2019), performs a masked atom prediction task on the molecular graph, predicting atom types such as carbon, hydrogen, oxygen, and nitrogen. Its variation, FG AttentiveFP, shares the same architecture as AttentiveFP, but it predicts both the atom type and the associated functional group. Experimental results indicate that incorporating functional group information significantly improves the model's performance on downstream tasks.

We also evaluate the BERT model trained on canonical SMILES strings, and its counterpart, FG BERT, which is trained on FG-enhanced SMILES. Results show that providing additional chemical context about functional groups boosts model performance in downstream tasks.

Finally, **FARM** (FG BERT with contrastive learning) integrates molecular structure representations from link prediction embeddings. **FARM** consistently achieves the highest performance across 6 out of 7 downstream tasks, demonstrating the power of combining FG-enhanced SMILES and contrastive learning.

Table 4 presents the detailed results of the aforementioned models across various MoleculeNet tasks, illustrating the performance of each architecture. For these experiments, we used random splitting to divide the downstream datasets into training, validation, and test sets in an 8:1:1 ratio. While random splitting is used consistently across models in this ablation study, scaffold splitting is applied for benchmarking to ensure a fair comparison with other methods.

### E.2  Fragmentation Methods

In this section, we compare the performance of our FG-based fragmentation method with BRICS fragmentation. When using BRICS as the fragmentation method and enriching SMILES with functional group information derived from BRICS (treating each BRICS fragment as a functional group), the resulting vocabulary size is 575,612 (with a minimum frequency of 5), compare to 14,741 with our FG-based fragmentation. This significant difference arises because BRICS tends to generate larger fragments, causing the vocabulary to grow rapidly. This larger vocabulary can make the model harder to train and converge, as it includes a high number of rare tokens. Consequently, the likelihood of encountering out-of-vocabulary tokens during validation and testing increases, which negatively impacts model performance. As shown in Table 5, under the same experimental setup, our FG-based fragmentation method outperforms BRICS fragmentation across all six MoleculeNet tasks.

## F  Benchmarking on ADMET Tasks

Table 6 presents the performance of FARM on the ADMET datasets. The ADMET leaderboard[2] evaluates model performance across ADMET tasks (Absorption, Distribution, Metabolism, Excretion, Toxicity) using a standardized train/validation/test split, which we adopt for consistency. FARM achieves state-of-the-art results on 3 out of 7 tasks and demonstrates competitive performance with other top models on the remaining tasks, highlighting its robustness in ADMET prediction. Notably, none of the foundation models for small molecules included in Table 2 have reported results on these ADMET tasks, primarily due to the limitations of the datasets themselves.

Many ADMET tasks in TDC are limited by small and highly imbalanced datasets, often containing fewer than 100 positive examples. These constraints make it difficult for models to learn meaningful, generalizable patterns, particularly

---

[2]https://tdcommons.ai/benchmark/admet_group/overview/

for complex endpoints like toxicity or clearance, which involve diverse and mechanistically unrelated biological pathways. For instance, a dataset labeling 1,000 compounds as "toxic" may include compounds that exert toxicity through entirely different mechanisms, preventing models from reliably capturing features that generalize to unseen cases. Consequently, performance on many TDC ADMET tasks is often near random, limiting their utility for assessing generalization. Specific tasks such as HIA_Hou, Clearance_Hepatocyte_AZ, and CYP3A4_Substrate_CarbonMangels exemplify this limitation, where even top-performing models fail to achieve meaningful results. To mitigate this, we evaluated FARM on a subset of more reliable TDC ADMET tasks, comparing its performance against all leading models, including some whose code is not publicly runnable (e.g., MapLight models). Many leaderboard approaches rely heavily on hand-crafted features and complex pipelines, which are orthogonal to our focus on end-to-end, transferable representation learning; combining FARM with such strategies presents a promising direction for future work.

## G  Benchmarking on Photostability

To contextualize FARM's performance on the D-B-A oligomer photostability dataset, we compare against two representative baselines: (i) a traditional machine learning model based on Morgan fingerprints and Random Forest (RF), and (ii) MolCLR, a graph contrastive learning framework that learns molecular representations through self-supervised pretraining. All methods are evaluated under the same hard-case scaffold split and protocol described in Section 5. In this split, 3 donor-bridge blocks and 15 acceptor blocks are held out entirely, ensuring that no molecule in the validation or test sets shares a donor or acceptor block with any training molecule. The Morgan fingerprint + RF baseline provides a strong descriptor-based reference, while MolCLR serves as a learned graph representation baseline. Together, these comparisons allow readers to assess whether the gains achieved by FARM arise from its functional-group-aware molecular representation rather than from general molecular descriptors or existing self-supervised graph learning approaches.

Table 11: Performance comparison of MolCLR, Morgan fingerprint + Random Forest (RF), and FARM on the D-B-A oligomer photostability dataset. $R^2$ scores are reported with standard deviation over 10 runs. The last two columns show the performance gain of FARM over RF and MolCLR, respectively.

| Task | MolCLR $R^2$ | RF $R^2$ | FARM $R^2$ | $\Delta R^2$ (FARM-RF) | $\Delta R^2$ (FARM-MolCLR) |
|---|---|---|---|---|---|
| HOMOm1 (eV) | $0.8607 \pm 0.01$ | $0.7878 \pm 0.0121$ | $\mathbf{0.927 \pm 0.015}$ | $+0.1392$ | $+0.0663$ |
| HOMO (eV) | $0.7752 \pm 0.01$ | $0.7967 \pm 0.0129$ | $\mathbf{0.895 \pm 0.030}$ | $+0.0983$ | $+0.1198$ |
| LUMO (eV) | $0.7131 \pm 0.01$ | $0.7530 \pm 0.0159$ | $\mathbf{0.928 \pm 0.009}$ | $+0.1750$ | $+0.2149$ |
| LUMOp1 (eV) | $0.5683 \pm 0.04$ | $0.5645 \pm 0.0386$ | $\mathbf{0.922 \pm 0.028}$ | $+0.3575$ | $+0.3537$ |
| PrimeExcite (eV) | $0.4922 \pm 0.02$ | $\mathbf{0.9432 \pm 0.0068}$ | $0.877 \pm 0.049$ | $-0.0662$ | $+0.3848$ |
| PrimeExcite (osc) | $0.7726 \pm 0.00$ | $0.2296 \pm 0.0335$ | $\mathbf{0.837 \pm 0.024}$ | $+0.6074$ | $+0.0644$ |
| DipoleMoment (D) | $0.5609 \pm 0.03$ | $\mathbf{0.7874 \pm 0.0555}$ | $0.755 \pm 0.053$ | $-0.0324$ | $+0.1941$ |
| LogP | $0.9581 \pm 0.01$ | $0.9357 \pm 0.0032$ | $\mathbf{0.995 \pm 0.006}$ | $+0.0593$ | $+0.0369$ |
| T1 | $0.7155 \pm 0.01$ | $0.7307 \pm 0.0105$ | $\mathbf{0.929 \pm 0.006}$ | $+0.1983$ | $+0.2135$ |
| T2 | $0.7657 \pm 0.01$ | $0.6375 \pm 0.0185$ | $\mathbf{0.925 \pm 0.018}$ | $+0.2875$ | $+0.1593$ |
| T3 | $0.5952 \pm 0.04$ | $0.7276 \pm 0.0349$ | $\mathbf{0.900 \pm 0.010}$ | $+0.1724$ | $+0.3048$ |
| T4 | $0.5690 \pm 0.03$ | $0.7955 \pm 0.0363$ | $\mathbf{0.921 \pm 0.020}$ | $+0.1255$ | $+0.3520$ |
| T5 | $0.6302 \pm 0.03$ | $0.6201 \pm 0.0283$ | $\mathbf{0.905 \pm 0.032}$ | $+0.2849$ | $+0.2748$ |
| S1 | $0.7067 \pm 0.01$ | $0.7273 \pm 0.0127$ | $\mathbf{0.936 \pm 0.016}$ | $+0.2087$ | $+0.2293$ |
| S2 | $0.7900 \pm 0.01$ | $0.6436 \pm 0.0180$ | $\mathbf{0.948 \pm 0.016}$ | $+0.3044$ | $+0.1580$ |
| S3 | $0.6217 \pm 0.02$ | $0.8083 \pm 0.0267$ | $\mathbf{0.911 \pm 0.008}$ | $+0.1027$ | $+0.2893$ |
| S4 | $0.7361 \pm 0.00$ | $0.8665 \pm 0.0107$ | $\mathbf{0.919 \pm 0.008}$ | $+0.0525$ | $+0.1829$ |
| S5 | $0.6412 \pm 0.02$ | $0.7592 \pm 0.0216$ | $\mathbf{0.924 \pm 0.014}$ | $+0.1648$ | $+0.2828$ |
| O1 | $0.7793 \pm 0.02$ | $0.5039 \pm 0.0243$ | $\mathbf{0.839 \pm 0.034}$ | $+0.3351$ | $+0.0597$ |
| TDOS4.0 | $0.7880 \pm 0.01$ | $0.8491 \pm 0.0075$ | $\mathbf{0.919 \pm 0.031}$ | $+0.0699$ | $+0.1310$ |
| **Average** $R^2$ | $0.7025$ | $0.7381$ | $\mathbf{0.9116}$ | $+0.1735$ | $+0.2091$ |

FARM achieves the best performance among the three methods on the majority of tasks, outperforming both the Morgan fingerprint + RF baseline and MolCLR on 18 of the 20 prediction tasks. The largest gains over MolCLR are observed for properties associated with excited-state and transition energies, including PrimeExcite (eV) (+0.385 $R^2$), LUMOp1 (+0.354), T4 (+0.352), T3 (+0.305), and S3 (+0.289). These results suggest that FARM's functional-group-aware representation is particularly effective at capturing the long-range structural and electronic patterns underlying excited-state behavior, which are difficult to model using fixed fingerprints or generic graph contrastive learning objectives.

The two tasks where FARM does not achieve the highest $R^2$ score are PrimeExcite (eV) and DipoleMoment, where the RF baseline remains slightly superior. Notably, these are also among the lowest-performing tasks across all methods. This observation suggests that these properties may depend more strongly on subtle electronic effects, molecular conformations, or three-dimensional structural information that are not fully encoded by either molecular fingerprints

or current 2D graph-based representations. Nevertheless, FARM remains competitive on these challenging tasks while providing substantial improvements on most other photophysical properties.

## H   Computational Resources

For the BERT model, even with an expanded vocabulary of approximately 14,000 tokens, the model size remains smaller than the English BERT vocabulary and does not require additional computational resources compared to training a standard BERT model. Each BERT model, with around 100 million parameters, was trained for 20 epochs on a machine with an NVIDIA A100 GPU (40GB memory), 256GB of RAM, and a 32-core AMD EPYC 7502 CPU. Training each model took approximately 6 hours. The experiments were conducted on an internal computing cluster, and no additional compute was required beyond the primary experiments included in the paper.

For other models, such as those used for ComplEx and link prediction GNN, the computational requirements are relatively lightweight, and the same machine used for BERT training was sufficient without the need for additional resources.

