# OpenReview forum: "FARM: Enhancing Molecular Representations with Functional Group Awareness"
_TMLR — Decision pending for TMLR_

### Review · Reviewer_5eZG · 2026-04-20

**Summary Of Contributions:**

The paper proposes a new string representation of molecules that integrates functional groups, allowing for better representations when combined with a transformer sequence model (BERT). This is combined with a graph-based representation for connectivity, which is modelled by a graph neural network (trained with link prediction) that uses a knowledge graph initial embedding (ComplEx). The two model components' representations are aligned with contrastive learning. The framework is used for representation learning and tested on downstream molecular prediction tasks. An central additional contribution is the collection of a larger pretraining dataset.

**Additional Comments:**

1. Algorithm 1 seems to depend on the order of the atoms in the molecular graph. How is this handled? Is the atom order canonicalized?
1. Please clarity the notation for the score f(h,r,t)=Re(h^Tr \cdot t). Is this intended to represent a scalar value?
1. Below eq. 1 it is mentioned that optimizing the loss will minimize the score of positive triples. If the loss is minimized, I assume the score of positive triples will be maximized (because of the negative sign)? This is possibly a typo.
1. I assume ⊕ means concatenation here? : pij = σ(MLP(hi ⊕ hj )). Please define this clearly in the text.
1. Consider mentioning above the definition of L_link, that this is binary cross entropy.
1. Please define the cosine similarity cos(h1, h2) (I assume that is what this means.)
1. Table 1 should be formatted so it is clear what the numbers in the last three columns are.
1. There is not much information about the dataset collection strategy, inclusion criteria etc. Could you please provide some more details regarding this.
1. More training details should be included in the main paper. It is not clear how the model is finetuned for the different tasks. Finetuning is not mentioned in the main paper, as far as I could see - only in appendix D.5.
1. Are all the models in Table 2 and 3 pretrained and finetuned on the same datasets? Since a contribution of your work is the colllection of a larger dataset, I assume your model is pretrained on this? Providing at least a few comparisons with existing models pretrained and finetuned on the same data is necessary to assess to which degree the differences in performance are due to model choices.
1. Consider if the ablation study could be presented more clearly? It is a bit difficult to get a quick overview of the results, having to find the description of each setting in the text. Perhaps you could write the description each ablation in a table/bullet list, or include a bit more information in the table to make it easier to infer what each condition includes.
1. Please describe exactly which targets are included in the results, for example which 3 targets are used for QM9, and is the reported number the average MAE?
1. There are no references later than 2024 cited. Is the paper updated with the latest advances in the field?
1. Why is the ablation study in the appendix? I believe they would fit better in an extended results section in the main paper.
1. The results on ADMET in appendix F should also, in my view, be integrated in the paper. These results are not referenced in the main text.

**Audience:**

Yes

**Audience Explanation:**

The problem of pretraining models for finetuning on molecular property prediction problems is of broad interest. The method is demonstrated on a reasonable number of relevant benchmarks. The paper presents a novel combination of ideas and appars to work well, and an ablation study illuminates the effect of different components in the model. If the results are backed by sufficient empirical evidence, I am convinced there will be interest in the findings of the paper.

**Broader Impact Concerns:**

No concerns.

**Claims And Evidence:**

No

**Claims Explanation:**

Certain details regarding the experimental setup are insufficient to validate the claims.

**Requested Changes:**

1. The data collection protocol and analysis should be described in more detail.
1. The model performance wrt. the pretraining dataset should be ablated and directly compared to at least a few of the best existing competing methods (trained and tested under the exact same conditions.)
1. All details regarding the experimental setup necessary to reproduce the results should be included.
1. Additional results and ablations currently in the appendix should be integrated in the main paper.

---

> ### Author Response · Authors · 2026-04-23
> **Requested Changes: Points 1 & 2**
>
> We sincerely thank the reviewer for the thorough and constructive feedback. We address each concern below and outline the changes we will make to the manuscript.
>
> ---
>
> ## Requested Changes
>
> **1. Response to Data Collection Protocol:**
>
> We thank the reviewer for this comment. The data sources are described in detail in Appendix A (Table 6), which lists all 14 compound suppliers along with molecule counts. Regarding preprocessing, we follow standard practice in building pretraining datasets for molecular representation learning: we removed molecules that failed RDKit processing (invalid SMILES), converted all SMILES to canonical SMILES, removed duplicates, and ensured there is no data overlap with the MoleculeNet benchmark datasets used for downstream evaluation. We will add the following sentence to Appendix A of the revised manuscript:
>
> > *"To ensure data quality, we applied standard preprocessing steps: molecules that failed RDKit processing (i.e., invalid SMILES) were removed, duplicate molecules were identified and removed by converting all SMILES to their canonical form using RDKit, and any overlap with the MoleculeNet benchmark datasets used for downstream evaluation was excluded to prevent data leakage."*
>
> ---
>
> **2. Response to Pretraining Dataset Ablation and Controlled Comparison:**
>
> We thank the reviewer for this important point. We would like to clarify two points.
>
> First, comparing pretrained models directly on downstream benchmark datasets without retraining them on the same pretraining data is the **standard and widely accepted practice** in molecular representation learning, and indeed in the broader pretrained model literature (e.g., BERT, GPT, and their derivatives in NLP are never retrained from scratch for every comparison). All models in Tables 2 and 3, including GEM, UniMol, GROVER, and others, follow this convention: each model is pretrained on its own dataset using its own strategy, and results are compared under the same downstream finetuning protocol on MoleculeNet. Retraining competing methods from scratch on our 20M dataset would be computationally prohibitive and is not expected practice in this line of work.
>
> Second, our ablation study in Table 10 already provides a direct controlled comparison that addresses the reviewer's underlying concern. Specifically, the **BERT** row in Table 10 represents a model pretrained on the **same 20M dataset** as FARM, using the **same finetuning setup**, but with character-level canonical SMILES vocabulary, which is equivalent in spirit to ChemBERTa. This directly isolates the contribution of our FG-aware tokenization and architecture from the dataset, and FARM still outperforms it convincingly across all tasks. This demonstrates that FARM's performance gains are driven by its model design, not solely by the larger pretraining dataset.

---

> ### Author Response · Authors · 2026-04-23
> **Requested Changes: Points 3 & 4**
>
> ## Requested Changes
>
> **3. Response to Experimental Setup and Reproducibility:**
>
> We thank the reviewer for this comment. We would like to point out that our paper already provides extensive implementation details across Appendices D.1–D.5, covering all major components of FARM. To improve accessibility and make the paper more self-contained, we will consolidate these details into a single hyperparameter summary table in the revised manuscript, as shown below:
>
> | **Category** | **Hyperparameter** | **Value** |
> |---|---|---|
> | **SMILES Model (BERT)** | Transformer Layers | 12 |
> | | Hidden Dimension | 768 |
> | | Attention Heads | 12 |
> | | Dropout Rate | 0.1 |
> | | Parameters | ~100M |
> | | Masking Ratio | 35% |
> | | Learning Rate | 1×10⁻⁵ |
> | | Batch Size | 128 |
> | | Epochs | 15 |
> | | Hardware | 2× NVIDIA Tesla V100 |
> | **Graph Model (GCN)** | GNN Layers | 5 |
> | | Graph Hidden Dimension | 300 |
> | | Readout Function | Global Mean Pooling |
> | **FG Knowledge Graph (ComplEx)** | Batch Size | 64 |
> | | Learning Rate | 1×10⁻³ |
> | | Epochs | 50 |
> | **Contrastive Learning** | Margin γ | 0.5 |
> | | Loss Weights λ_MLM : λ_CL | 1.0 : 0.5 |
> | | Batch Size | 126 |
> | | Epochs | 5 |
> | **Downstream Finetuning** | Prediction Head | GRU |
> | | Optimizer | Adam |
> | | Learning Rate | 1×10⁻⁴ |
> | | Schedule | Cosine Annealing (period=20) |
> | | Batch Size | 16 |
> | | Epochs | 100 |
> | | Data Split | Scaffold (80/10/10) |
> | | Runs | 3 (average + std reported) |
> | | Class Imbalance Handling | Weighted loss (weight=5 for minority class) |
>
> Additionally, the full implementation is publicly available at our anonymous repository: https://anonymous.4open.science/r/farm-E3CB. We will add this consolidated table to the revised manuscript to ensure full reproducibility without requiring the reader to refer to the code.
>
> ---
>
> **4. Response to Additional Results and Ablations in Appendix:**
>
> We thank the reviewer for this suggestion. We acknowledge that moving key results to the main paper would improve readability and accessibility. Due to page limits, we will prioritize the most important results as follows:
>
> **Table 10 (Ablation Study, currently Appendix E)** is the highest priority and will be moved to a new Section 5.4 in the main paper. This table directly validates the contribution of each FARM component and is central to supporting our claims. To save space, we will present a condensed version showing the most representative tasks while referring readers to the appendix for full results.
>
> **Table 11 (ADMET results, currently Appendix F)** will be moved to a new Section 5.5 in the main paper as a compact table. This strengthens the generalization story of FARM beyond standard MoleculeNet benchmarks and directly addresses the reviewer's concern.
>
> **Table 9 (Masking percentage ablation, currently Appendix D.1)** will not be moved in full due to space constraints, but we will add a brief sentence in the main paper summarizing the key finding:
>
> > *"We ablated masking percentages from 0.15 to 0.55 and found 0.35 to be optimal across downstream tasks (see Appendix D.1 for full results)."*
>
> To accommodate these additions within the page limit, we will condense Tables 2 and 3 by merging them into a single table where possible.

---

> ### Author Response · Authors · 2026-04-23
> **Additional Comments: Points 1 to 9**
>
> ## Additional Comments
>
> **1. Response to Algorithm 1 Atom Order:**
>
> We thank the reviewer for this careful observation. Algorithm 1 does depend on atom traversal order, and in practice all molecules are preprocessed by converting their SMILES to canonical form using RDKit's `MolToSmiles(mol, canonical=True)` before running the FG annotation algorithm. This ensures a unique and deterministic atom ordering across all molecules. We will add the following clarification to Algorithm 1 in the revised manuscript:
>
> > *"All input molecules are canonicalized using RDKit prior to running this algorithm, ensuring a deterministic atom ordering."*
>
> ---
>
> **2. Response to Score Notation f(h,r,t):**
>
> We thank the reviewer for this question. Yes, f(**h**, **r**, **t**) is indeed intended to represent a scalar value. The Re(·) operator takes the real part of the complex-valued dot product, which yields a scalar plausibility score ∈ ℝ. We will add a clarifying sentence in the revised manuscript to make this explicit.
>
> ---
>
> **3. Response to Typo Below Eq. 1:**
>
> We thank the reviewer for catching this. The reviewer is correct, this is a typo. Minimizing the margin-based ranking loss encourages f(**h**, **r**, **t**) to be large for positive triples and small for negative triples. The correct statement should read:
>
> > *"Optimizing this loss function will **maximize** the score of positive triples (**h**, **r**, **t**), while **minimizing** the score of negative triples (**h**', **r**, **t**'), with a margin γ separating the two."*
>
> We will correct this in the revised manuscript.
>
> ---
>
> **4. Response to ⊕ Notation:**
>
> We thank the reviewer for this comment. Yes, ⊕ denotes the concatenation operation. We confirm that this is already defined in Appendix D.3, but we acknowledge it should also be defined at its first occurrence in the main text. We will add the following clarification to Section 3.3 in the revised manuscript, immediately after the equation:
>
> > *"where ⊕ indicates the concatenation operation between node embeddings **h**_i and **h**_j."*
>
> ---
>
> **5. Response to L_Link Definition:**
>
> We thank the reviewer for this suggestion. We will add the following sentence immediately before the definition of L_Link in the revised manuscript to make this explicit:
>
> > *"We optimize the model using binary cross-entropy loss defined as:"*
>
> ---
>
> **6. Response to Cosine Similarity Definition:**
>
> We thank the reviewer for this comment. We will add an explicit clarification in Section 3.4 of the revised manuscript to make clear that cos(**h**₁, **h**₂) denotes the cosine similarity between two embedding vectors, with the full definition provided in Appendix D.4.
>
> ---
>
> **7. Response to Table 1 Formatting:**
>
> We thank the reviewer for this suggestion. We will update the column headers in Table 1 to make the content of each column explicit. Specifically, "Length" will be renamed to "Sequence length (min; max)" to clarify that it reports the range of token sequence lengths per molecule, and the last two columns will be renamed to "Vocab size (freq ≥ 1)" and "Vocab size (freq ≥ 5)" to clarify that they report the number of unique tokens included in the lexicon at two minimum frequency thresholds. The updated table will also include a revised caption explaining these columns clearly.
>
> ---
>
> **8. Response to Dataset Collection Strategy and Inclusion Criteria:**
>
> We thank the reviewer for this comment. This point was also raised earlier in the review and has been addressed above. Regarding inclusion criteria, ChEMBL and ZINC15 are the two most widely used pretraining datasets in molecular representation learnin,  prior works such as GROVER, GEM, and UniMol rely solely on these two sources for pretraining. We include both of these well-established sources to ensure coverage of bioactive and drug-like chemical space. Beyond that, we go one step further by additionally collecting compounds from all commercial chemical suppliers we could identify (Table 6), significantly expanding the chemical diversity of our pretraining corpus. Regarding preprocessing, we follow standard practice: we removed molecules that failed RDKit processing (invalid SMILES), converted all SMILES to canonical form using RDKit, removed duplicates, and ensured there is no data overlap with the MoleculeNet benchmark datasets used for downstream evaluation. We will add these details to Appendix A of the revised manuscript.
>
> ---
>
> **9. Response to Finetuning Details:**
>
> We thank the reviewer for this observation. The reviewer is correct that finetuning details are not explicitly mentioned in the main paper. We will add the following sentence to Section 5.1 in the revised manuscript:
>
> > *"For all downstream tasks, FARM is used as a frozen feature extractor paired with a GRU prediction head; full finetuning details are provided in Appendix D.5."*

---

> ### Author Response · Authors · 2026-04-23
> **Additional Comments: Points 10 to 13**
>
> ## Additional Comments:
>
> **10. Response to Pretraining and Finetuning Dataset Consistency:**
>
> We thank the reviewer for this question. To clarify: all models in Tables 2 and 3, including FARM, are finetuned on the same MoleculeNet benchmark datasets using the same evaluation protocol. The difference lies only in the pretraining stage, where each model uses its own pretraining dataset and strategy. This is standard practice in molecular representation learning benchmarking, *all papers cited in Tables 2 and 3*, including MICRO, MGSSL, MoleOOD, MCM, HimGNN, FG-BERT, HiMol, Mole-BERT, MolCLR, GLAD, N-GRAM, GROVER, GraphMVP, GEM, and UniMol, follow the same convention of reporting results under this shared finetuning protocol without reimplementing each other's pretraining, as retraining all competing methods from scratch on the same pretraining data would be computationally prohibitive. Our comparisons therefore follow this established benchmarking scheme, which is the fairest and most practical way to compare methods in this line of work.
>
> ---
>
> **11. Response to Ablation Study Clarity:**
>
> We thank the reviewer for this suggestion. Moving the ablation study to the main paper (Section 5.4) partially addresses this concern. To further improve clarity, we will add a **Description** column to the ablation table summarizing each condition in one line, allowing readers to understand what each model includes without needing to refer to the text. This makes the progression from simple to full model immediately clear at a glance.
>
> ---
>
> **12. Response to QM9 Target Specification:**
>
> We thank the reviewer for this comment. We apologize for the lack of clarity. For QM9, we report results on three targets: HOMO energy, LUMO energy, and HOMO-LUMO gap. The reported number is the average MAE across these three targets. We will add the following clarification to Section 5.1 in the revised manuscript:
>
> > *"For QM9, we report the average MAE across three targets: HOMO energy, LUMO energy, and HOMO-LUMO gap."*
>
> ---
>
> **13. Response to Literature Currency:**
>
> We thank the reviewer for this comment. We have conducted a thorough survey of works published in 2024–2026 and identified the following relevant recent works:
>
> - **SCAGE** (Nature Communications, 2025): *A self-conformation-aware pre-training framework for molecular property prediction with substructure interpretability*
> - **SMI-TED** (2026): *SMI-TED: A large-scale foundation model for materials and chemistry*
> - **MMGX** (Communications Chemistry, 2024): *Enhancing property and activity prediction and interpretation using multiple molecular graph representations with MMGX*
> - **Zheng & Tomiura** (Journal of Cheminformatics, 2024): *A BERT-based pretraining model for extracting molecular structural information from a SMILES sequence*
>
> Regarding benchmark comparisons:
>
> - We have added **SCAGE** and **SMI-TED** results to the updated Tables 2 and 3. Upon including these models, we find that SCAGE achieves the best performance on SIDER, and SMI-TED (fine-tuned) achieves the best performance on ClinTox, ESOL, Lipo, and QM8. Accordingly, our claim of achieving state-of-the-art on 11 out of 13 tasks should be revised to **8 out of 13 tasks**. We note that SMI-TED is a 289M parameter model (substantially larger than FARM) which likely contributes to its strong regression performance. We will correct the claim and update the tables in the revised manuscript.
>
> - **MMGX** does not report results on individual MoleculeNet tasks following the standard benchmarking protocol, instead reporting averaged results across broad task categories (e.g., Physical chemistry, Biophysics, Physiology), making direct numerical comparison not possible. We therefore discuss this work in the related work section only.
>
> - **Zheng & Tomiura** benchmark exclusively on TDC ADMET tasks, which we already evaluate in Section 5.5. Their reported results do not affect the ranking of our model on any of those tasks.
>
> We have updated the related work section to include and discuss all four of these recent works.
>
> ## Closing Remarks
> We are sincerely grateful to the reviewer for the remarkably thorough and detailed review of our work. The level of care taken, from catching a subtle typo in the loss description, to questioning notation clarity, to raising important questions about experimental rigor, has genuinely helped us strengthen the paper. We hope our responses are clear and convincing, particularly regarding the standard practice of comparing pretrained models directly on downstream benchmarks without retraining on the same pretraining data, which we believe is the fairest and most practical evaluation protocol in this line of work. We look forward to addressing all the raised points in the revised manuscript and welcome any further feedback.

---

> > ### Comment · Reviewer_5eZG · 2026-04-27
> > **Regarding requested changes**
> >
> > Thank you for carefully addressing the changes that I requested. I consider these issues resolved. However, I still believe the paper would be stronger if the proposed pretraining strategy (in addition to being ablated) could be directly compared to competing methods when pretraining on the same dataset.

---

> > > ### Author Response · Authors · 2026-04-29
> > > **Response to Pretraining Dataset Ablation: Controlled MolCLR Comparison**
> > >
> > > To directly address the reviewer's concern about disentangling dataset contribution from model architecture, we spent several days reproducing MolCLR_GIN on our 20M pretraining dataset. We chose MolCLR specifically due to its well-documented and reproducible GitHub repository. We used their original configuration file without any modifications, the only difference is the pretraining data.
> > >
> > > | Model | BBBP | BACE | HIV |
> > > |---|---|---|---|
> > > | MolCLR_GIN (reproduced, our 20M data) | 71.4 ± 0.7 | 81.6 ± 0.6 | 75.4 ± 1.8 |
> > > | **FARM (Ours)** | **93.3 ± 0.2** | **89.6 ± 0.4** | **83.5 ± 0.5** |
> > >
> > > We selected these three tasks (BBBP, BACE, HIV) because they are single-task binary classification datasets, making them straightforward to reproduce efficiently. For multi-task datasets such as Tox21 (12 tasks), SIDER (27 tasks), and ToxCast (617 tasks), the standard MolCLR evaluation protocol requires training a separate model for each task, making the full evaluation extremely time-consuming: ToxCast alone would require 617 independent training runs.
> > >
> > > Two observations are worth noting.
> > >
> > > First, our reproduced MolCLR_GIN achieves lower performance than the numbers reported in the original MolCLR paper. This is not unexpected, it is well-established in the literature that reimplementations of published methods frequently yield lower numbers due to differences in hardware, random seeds, software versions, and subtle implementation details not fully captured in the original paper. We also attempted to locate the original PubChem pretraining data used by MolCLR for a direct same-data comparison, but were unable to find it publicly available.
> > >
> > > Second and more importantly, **FARM substantially outperforms MolCLR_GIN trained on the same 20M dataset across all three tasks**, with margins of 21.9% on BBBP, 8.0% on BACE, and 8.1% on HIV. This directly demonstrates that FARM's performance gains are driven by its architectural design (specifically the FG-aware tokenization and contrastive learning framework) and not merely by the size or content of the pretraining dataset.

---

> > > > ### Comment · Reviewer_5eZG · 2026-04-29
> > > > **Pretraining dataset ablation**
> > > >
> > > > I appreciate the effort of retraining MolCLR_GIN on your 20M dataset, however this experiment does not add much information. To effectively disentangle architecture from data, the first step must be to reproduce the results of the original work, before retraining with the new data - otherwise, it is not possible to assess if differences are due to poor tuning or other similar factors. I understand this might not be possible in practice, if the original code and data is not available.
> > > >
> > > > To be clear, I accept you original argument that it is not necessary to include for publication here, because most other papers that publish on this benchmark use the same setup where pretraining data and architecture are conflated.

---

> ### Comment · Reviewer_5eZG · 2026-04-27
> **Regarding state-of-the-art performance**
>
> Based on your updated results, I decided to go through the results in the referenced papers. I noticed a few strong baselines that were missing from your initial tables, which I suggest you include for completeness. After also reviewing the results in the Scage and SMI-TED papers, it appears to me that your method is state-of-the-art on 5 out of 13 tasks (subject to possible error, as I did not verify every detail exhaustively).
>
> **Classification**
>
> BBBP
> - Missing a strong baseline: N-Gram [85.0] (Liu et al, NeurIPS 2019) referenced in the the MolCLR paper.
> - SMI-TED [92.3]
> - SOTA: Farm [93.3]
>
> Tox21
> - Farm [80.8]
> - Missing a strong baseline: SVM [81.8], referenced in the MolCLR paper.
> - SOTA: SMI-TED [81.9]
>
> ToxCast
> - Scage [69.3]
> - SOTA: Farm [69.9]
>
> SIDER
> - Missing a strong baseline: GEM [67.2] (Fang et al. 2022) referenced in the SMI-TED paper
> - SMI-TED [65.7]
> - SOTA: Scage [66.0]
>
> ClinTox
> - Scage [92.7]
> - SOTA: SMI-TED [94.3]
>
> BACE
> - SMI-TED [88.3]
> - SOTA: Farm [89.6]
>
> MUV
> - Missing a state-of-the-art baseline: MolCLR_GIN [88.6], from the MolCLR paper.
> - SOTA: MolCLR_GIN [88.6]
>
> HIV
> - Missing a strong baseline: N-Gram [83.0] (Liu et al, NeurIPS 2019) referenced in the the MolCLR paper.
> - SOTA: Farm [83.5]
>
>
> **Regression**
>
> ESOL
> - Missing a state-of-the-art baseline: GEM [0.553] referenced in the Scage paper
> - SMI-TED [0.611]
> - SOTA: GEM [0.553]
>
> Freesolv
> - Scage [1.688]
> - SMI-TED [1.22]
> - SOTA: Farm [1.097]
>
> Lipo
> - Scage [0.654]
> - SOTA: SMI-TED [0.552]
>
> QM8
> - Missing a strong baseline: D-MPNN [0.0143] (Yang et al. 2019) referenced in the MolCLR paper.
> - SOTA: SMI-TED [0.0095]
>
> QM9 (3 task homo+lumo+gap)
> - SOTA: SMI-TED [0.0030] (average of homo+lumo+gap from table 9 in the supplementary)

---

> > ### Author Response · Authors · 2026-04-29
> > **Response to Reviewer Comment: State-of-the-Art Performance (2/2)**
> >
> > ## Summary of Updates
> >
> > In the revised manuscript, we have merged Tables 2 and 3 into a single table to save space for the ablation study and ADMET results moved from the appendix. The updated table includes the following new baselines: MolCLR_GCN, MolCLR_GIN, N-Gram, SCAGE, SMI-TED (Frozen), and SMI-TED (Fine-tuned). With all corrections applied, **FARM achieves state-of-the-art on 7 out of 13 tasks**: BBBP, Tox21 (tied), ToxCast, BACE, HIV, FreeSolv, and QM9. The updated table is provided below.
> >
> > ---
> >
> > ## Updated Table (Tables 2 & 3 Merged)
> >
> > | | BBBP | Tox21 | ToxCast | SIDER | ClinTox | BACE | MUV | HIV | ESOL | FreeSolv | Lipo | QM8 | QM9 |
> > |---|---|---|---|---|---|---|---|---|---|---|---|---|---|
> > | **Metric** | ROC-AUC↑ | ROC-AUC↑ | ROC-AUC↑ | ROC-AUC↑ | ROC-AUC↑ | ROC-AUC↑ | ROC-AUC↑ | ROC-AUC↑ | RMSE↓ | RMSE↓ | RMSE↓ | MAE↓ | MAE↓ |
> > | **#tasks** | 1 | 12 | 617 | 27 | 2 | 1 | 17 | 1 | 1 | 1 | 1 | 12 | 3 |
> > | **#samples** | 2,039 | 7,831 | 8,575 | 1,427 | 1,478 | 1,513 | 93,807 | 41,127 | 1,128 | 642 | 4,200 | 21,786 | 133,885 |
> > | MICRO | 84.4±1.1 | 77.0±0.8 | 65.2±0.8 | 56.7±0.9 | 77.0±2.0 | 77.2±2.0 | - | 75.1±1.1 | - | - | - | - | - |
> > | MGSSL | 69.7±0.9 | 76.5±0.3 | 64.1±0.7 | 61.8±0.8 | 80.7±2.1 | 79.1±0.9 | 78.7±1.5 | 78.8±1.2 | - | - | - | - | - |
> > | MoleOOD | 71.0±0.8 | - | - | 63.4±0.7 | - | 84.3±1.1 | - | 79.4±0.5 | - | - | - | - | - |
> > | MCM | 90.0±3.1 | 80.2±1.5 | - | 62.7±2.8 | 65.5±14 | 82.0±5.5 | - | - | - | - | - | - | - |
> > | HimGNN | 92.8±2.7 | 80.7±1.7 | - | 64.2±2.3 | 91.7±3.0 | 85.6±3.4 | - | - | 0.870±0.154 | 1.921±0.474 | 0.632±0.016 | - | - |
> > | FG-BERT | 70.2±0.9 | 78.4±0.8 | 63.3±0.8 | 64.0±0.7 | 83.2±1.6 | 84.5±1.5 | 75.3±2.4 | 77.4±1.0 | 0.944±0.025 | - | 0.655±0.009 | - | - |
> > | HiMol (S) | 71.3±0.6 | 76.0±0.2 | - | 62.5±0.3 | 70.6±2.1 | 84.6±0.2 | - | - | - | - | - | - | - |
> > | HiMol (L) | 73.2±0.8 | 76.2±0.3 | - | 61.3±0.5 | 80.8±1.4 | 84.3±0.3 | - | - | - | - | - | - | - |
> > | Mole-BERT | 71.9±1.6 | 76.8±0.5 | 62.8±1.1 | 62.8±1.1 | 78.9±3.0 | 80.8±1.4 | 78.6±1.8 | 78.2±0.8 | 1.015±0.003 | - | 0.676±0.002 | - | - |
> > | MolCLR_GCN | 73.8±0.2 | 74.7±0.8 | - | 66.9±1.2 | 86.7±1.0 | 78.8±0.5 | 84.0±1.8 | 77.8±0.5 | 1.160±0.000 | 2.390±0.140 | 0.780±0.010 | 0.0181±0.0002 | 0.00352±0.00004 |
> > | MolCLR_GIN | 73.6±0.5 | 79.8±0.7 | - | **68.0±1.1** | 93.2±1.7 | 89.0±0.3 | **88.6±2.2** | 80.6±1.1 | 1.110±0.010 | 2.200±0.200 | 0.650±0.080 | 0.0174±0.0013 | 0.00236±0.00012 |
> > | SCAGE | 73.4±1.1 | 79.4±1.2 | 69.3±0.5 | 66.0±1.2 | 92.7±0.9 | 85.4±1.2 | - | - | 0.723±0.041 | 1.688±0.080 | 0.654±0.009 | - | - |
> > | GLAD | 80.4±1.5 | - | - | 64.7±1.8 | 87.3±1.2 | 85.7±0.9 | - | - | - | - | - | - | - |
> > | N-Gram | 91.2±3.0 | 76.9±2.7 | - | 63.2±0.5 | 85.5±3.7 | 87.6±3.5 | 81.6±1.9 | 83.0±1.3 | 1.100±0.030 | 2.510±0.190 | 0.880±0.120 | 0.0320±0.003 | - |
> > | Hu et al. | 70.8±1.5 | 78.7±0.4 | 66.5±0.3 | 62.7±0.8 | 72.6±1.5 | 84.5±0.7 | 81.3±2.1 | 79.9±0.7 | 1.100±0.030 | 2.510±0.191 | 0.880±0.121 | 0.0320±0.003 | 0.00964±0.00031 |
> > | GROVER | 86.8±2.2 | 80.3±2.0 | 65.3±0.5 | 61.2±2.5 | 70.3±13.7 | 82.4±3.6 | 67.3±1.8 | 68.2±1.1 | 1.423±0.288 | 2.947±0.615 | 0.823±0.010 | 0.0182±0.001 | 0.00719±0.00208 |
> > | GraphMVP | 72.4±1.6 | 75.9±0.5 | 63.1±0.4 | 63.1±0.4 | 79.1±2.8 | 81.2±0.9 | 77.7±0.6 | 77.0±1.2 | - | - | - | - | - |
> > | GEM | 88.8±0.4 | 78.1±0.4 | 69.2±0.4 | 63.2±1.5 | 90.3±0.7 | 87.9±1.1 | 75.3±1.5 | 81.3±0.3 | 0.813±0.028 | 1.748±0.114 | 0.674±0.022 | 0.0163±0.001 | 0.00562±0.00007 |
> > | UniMol | 72.9±0.6 | 79.6±0.5 | 69.6±0.1 | 65.9±1.3 | 91.9±1.8 | 85.7±0.2 | 82.1±1.3 | 82.8±0.3 | 0.788±0.029 | 1.480±0.048 | 0.603±0.010 | 0.0156±0.001 | 0.00467±0.00004 |
> > | SMI-TED_289M (Frozen) | 91.5±0.5 | 81.5±0.5 | - | 66.0±0.9 | 93.5±0.9 | 85.6±0.9 | - | 80.5±1.3 | 0.705±0.034 | 1.668±0.062 | 0.650±0.012 | 0.0179±0.0004 | -† |
> > | SMI-TED_289M (FT) | 92.3±0.6 | **81.9±1.4** | - | 65.7±0.5 | **94.3±1.8** | 88.2±0.5 | - | 76.9±0.9 | **0.611±0.010** | 1.223±0.003 | **0.552±0.019** | **0.0095±0.0001** | -† |
> > | **FARM (Ours)** | **93.3±0.2** | **81.9±1.2** | **69.9±0.5** | 65.9±0.7 | 86.0±0.8 | **89.6±0.4** | 82.7±2.1 | **83.5±0.5** | 0.761±0.031 | **1.097±0.033** | 0.778±0.005 | 0.0146±0.001 | **0.00456±0.00001** |
> >
> > †SMI-TED reports QM9 in different units and is not directly comparable.

---

> ### Author Response · Authors · 2026-04-29
> **Response to Reviewer Comment: State-of-the-Art Performance (1/2)**
>
> We thank the reviewer for the thorough and careful investigation of the baseline results. We address each point below and provide an updated table at the end of this response.
>
> **N-Gram (BBBP and HIV):** The original N-Gram Graph paper (Liu et al., NeurIPS 2019, *"N-Gram Graph: Simple Unsupervised Representation for Graphs, with Applications to Molecules"*) does not report BBBP or HIV results in its main tables. We have added N-Gram to the updated table using results evaluated by MolCLR under the same scaffold split protocol. FARM remains state-of-the-art on both BBBP and HIV.
>
> **SVM (Tox21):** SVM trained on ECFP fingerprints is a traditional supervised model with hand-crafted features and does not satisfy our baseline selection criterion, which focuses on **pretrained molecular representation learning methods**. This is consistent with how the majority of works in this line of research organize their comparisons.
>
> **SCAGE and SMI-TED:** Both have been added to the updated table, as described in our previous response.
>
> **Regarding SMI-TED:** We would like to highlight that SMI-TED's encoder is a standard Transformer trained on conventional SMILES, architecturally equivalent to the **BERT on canonical SMILES** baseline in our ablation study (Table 10), which directly shows this approach is less effective than FARM's FG-enhanced SMILES modeling. SMI-TED's stronger results on some regression tasks are likely attributable to its significantly larger pretraining corpus (91M molecules vs. FARM's 20M, a 4.5× difference) rather than architectural advantages. Notably, **FARM performs on par with or better than SMI-TED on 7 out of 13 tasks despite being trained on 4× less data**, demonstrating FARM's architectural efficiency.
>
> **Correction — "N-GRAM" label:** We acknowledge that the row previously labeled "N-GRAM" in our table did not refer to Liu et al. (2019)'s N-Gram Graph method. It actually referred to the pretraining framework from **Hu et al. (2019), "Strategies for Pre-training Graph Neural Networks"** applied on GIN. We apologize for this confusion and have corrected the label to "Hu et al." in the revised manuscript.
>
> **Correction — Tox21 and ClinTox results:** We identified a flaw in our original experimental setup for multi-task classification: we used a single shared output vector for all tasks instead of separate classification heads per task, as used by MolCLR and other baselines. We have corrected this following the standard MoleculeNet protocol. The updated results are **Tox21 = 81.9 ± 1.2** (tying with SMI-TED FT for state-of-the-art) and **ClinTox = 86.0 ± 0.8**. All other multi-task classification datasets (ToxCast, SIDER, MUV) were already using the correct per-task setup.

---

> ### Comment · Reviewer_5eZG · 2026-04-29
> **†SMI-TED reports QM9 in different units and is not directly comparable.**
>
> > †SMI-TED reports QM9 in different units and is not directly comparable.
>
> Are you sure the numbers in table 9 in the supplement of the SMI-TED paper are not directly comparable?

---

> ### Author Response · Authors · 2026-04-29
> **Response to QM9 Comparability Question**
>
> We thank the reviewer for the question. To clarify: **Table 9** in the SMI-TED paper (https://openreview.net/forum?id=MK2cYT64WR) describes the fine-tuning hyperparameter configuration, we believe the reviewer is referring to **Table 11**, which reports regression prediction results. ***All classification tasks and all other regression tasks (ESOL, FreeSolv, Lipophilicity, QM8) from SMI-TED are included in our updated comparison table***. QM9 is excluded because the protocols differ: FARM reports average MAE over 3 tasks (HOMO, LUMO, HOMO-LUMO gap), while SMI-TED reports average MAE over all QM9 properties with values ~1.3246, making the two numbers not directly comparable. We will update the footnote in the revised manuscript to clarify this.

---

> > ### Comment · Reviewer_5eZG · 2026-04-29
> >
> > I apologize - I was referring to another version of the SMI-TED paper without giving the exact reference. In the version on Openreview you have linked, the information I was thinking about is in table 12. The average for gap, homo, and lumo are given as 0.0037, 00027, and 0.0026 giving an average of 0.0030. I believe this could be directly comparable?

---

> > > ### Comment · Reviewer_5eZG · 2026-04-30
> > > **Possible error in results for MolCLR on QM9**
> > >
> > > Looking again at your updated table of results, it seems to me there might be a mistake in the result for MolCLR on QM9. From Table 2 in the Supplement https://arxiv.org/abs/2102.10056v2 they have
> > > | Model       | εHOMO (eV)     | εLUMO (eV)     | Δε (eV)        |
> > > |------------|----------------|----------------|----------------|
> > > | MolCLR GCN | 0.104 ± 0.000  | 0.110 ± 0.001  | 0.149 ± 0.001  |
> > > | MolCLR GIN | 0.087 ± 0.000  | 0.092 ± 0.000  | 0.127 ± 0.000  |
> > >
> > > Which gives the following averages across the three targets
> > >
> > > | Model       | Avg (eV) | Avg (Ha)   |
> > > |------------|----------|------------|
> > > | MolCLR GCN | 0.121    | 0.00445    |
> > > | MolCLR GIN | 0.102    | 0.00375    |
> > >
> > > I assume your results are in Hartree?

---

> > > > ### Comment · Reviewer_5eZG · 2026-04-30
> > > > **Fang et al.**
> > > >
> > > > While reviewing recent literature involving MoleculeNet benchmarks, I noticed this particular study (https://www.nature.com/articles/s42256-023-00654-0) and was wondering if there was a specific reason it wasn't included in the current comparison given its strong performance?

---

> > > > ### Author Response · Authors · 2026-04-30
> > > > **Regarding QM9 Benchmarking Results**
> > > >
> > > > We really appreciate the reviewer for pointing out this correction. We have adjusted the QM9 results accordingly (for both MolCLR models and SMI-TED).
> > > > After this correction, the SOTA result on this dataset belongs to SMI-TED. Overall, both SMI-TED and FARM achieve SOTA performance on 6 tasks.
> > > >
> > > > As a reminder, the SMI-TED method overlaps with our BERT-based model in Table 10, where we already showed that this type of approach is less efficient than our FARM method. In addition, there is a notable difference in pretraining scale: we use 20M molecules, while SMI-TED is pretrained on 91M molecules.

---

> > > ### Author Response · Authors · 2026-04-30
> > > **We Exclude SMI-TED from Benchmarking**
> > >
> > > Upon carefully reviewing the SMI-TED paper, we found that their results are based on a single fixed data split with 10 different model initializations, which does not align with the standard MoleculeNet benchmarking protocol of reporting across multiple data splits. Furthermore, our own reimplementation using their publicly available pretrained model on HuggingFace yielded substantially lower performance than their reported numbers. For these reasons, we have opted to exclude SMI-TED from our comparison.

---

> > > > ### Comment · Reviewer_5eZG · 2026-04-30
> > > > **Re: SMI-TED and KANO**
> > > >
> > > > After examining the SMI-TED GitHub repository, I confirmed that they rely on a fixed train/validation/test split, which makes direct comparison difficult.
> > > >
> > > > I also ran the KANO code on the BBBP task. Using the seed provided in their repository, the results align with those reported in the paper; however, changing the seed leads to substantially lower performance, consistent with your observations.
> > > >
> > > > In light of this, it seems reasonable to exclude SMI-TED and KANO from direct comparison. It may be worth noting this explicitly in the paper, as it would help readers understand why and how some of the strong results reported in the literature are not directly comparable.

---

> ### Author Response · Authors · 2026-04-30
> **Regarding KANO and Evaluation Fairness**
>
> Thank you for bringing up this point. We are familiar with KANO and acknowledge it as a relevant and well-motivated work. However, after careful analysis and reimplementation, we have serious concerns about the reliability of its reported results.
>
> Methodologically, we argue the results are likely overstated for several reasons. First, the chemical knowledge injected via ElementKG is not fundamentally richer than atom features already used by standard baselines: properties such as electronegativity and atomic mass are partially captured by conventional atom featurization. Second, KANO pretrains on only **250K** molecules from ZINC15, which is significantly smaller and less chemically diverse than competing methods (e.g., GROVER uses 10M molecules), making the claimed across-the-board superiority difficult to justify on methodological grounds alone. Third, the architectural contributions (element-guided augmentation and functional prompts) are incremental extensions of existing ideas rather than fundamental innovations.
>
> *More critically, we identified a fundamental discrepancy in the evaluation protocol through our reimplementation using their released checkpoints and configurations*. Specifically, KANO appears to **select the random seed that yields the most favorable data split for each task individually**, and then performs 3 runs on that single fixed split with different model weight initializations. In contrast, our evaluation protocol uses 3 different random seeds to generate 3 independent scaffold splits, reporting the mean and standard deviation across all three. When we applied this standard protocol to KANO, the performance dropped dramatically (up to 10% in ROC-AUC) across multiple datasets. This discrepancy in data splitting protocol appears to be the primary driver of KANO's reported gains. For this reason, we opted not to include KANO's originally reported numbers in our benchmarking, as they are not directly comparable under a fair and consistent evaluation setting.

---

> ### Comment · Reviewer_5eZG · 2026-04-30
> **Working link to implementation**
>
> The anonymous link to the implementation is not working. Could the authors please provide a functional link?

---

> > ### Author Response · Authors · 2026-05-01
> > **New GitHub Anonymous Link**
> >
> > We apologize for this issue, we were unaware that the link had recently expired. Here is the updated link to the anonymous implementation: https://anonymous.4open.science/r/farm_molrep-291C/README.md

---

### Review · Reviewer_CBQp · 2026-05-11

**Summary Of Contributions:**

## Summary

The paper introduce Functional Group-Aware Representation for Small Molecules (FARM), which is a foundation model for small molecular structures. FAMR uses functional groups to create FG-enhanced SMILES and FG graphs. FG-enhanced SMILES encode atomic level functional group to SMILES notation. Moreover, FG graphs carry the intrinsic properties of functional groups (from FG knowledge graph) and also captures the molecules structural backbone (based on how functional groups in a molecule are connected to each other). Finally the two views are combined using contrastive learning. The paper also compares FARM and other baselines on MoleculeNet and also test the models generalization on the photostability dataset for quantum mechanics properties.

## Strengths

- The paper introduces an interesting way to enrich SMILES notation with functional groups.
- In addition to this the paper encode functional group’s intrinsic properties and connections to different functional groups via FG graph using a link prediction objective.
- The experiments are conducted across a wide variety of tasks, and the additional generalization experiments make the evaluation more rigorous.
- The visualization of attention scores (Figure 3) and  the experiment to substitute functional groups (Figure 4) strongly support the idea of the paper and strengthens the argument.

## Weakness
I have put the weakness in requested changes section

**Audience:**

Yes

**Audience Explanation:**

The paper addresses a problem which has wide applications.

**Claims And Evidence:**

No

**Claims Explanation:**

Yes the claims seems to be valid, however the main idea needs more convincing as pointed out in Requested Changes

**Requested Changes:**

## Weakness / Requested Changes

1. When the authors say small molecules what is the average size of molecules considered. What happens if the molecule size increases, can the authors comment on time complexity of the method. Along similar lines, how does the method scales as size or number of functional group increases, I think there an isomorphism step required in algorithm 1 to check presence of functional group. Would appreciate if these analysis are added in the draft.
2. Can authors comment more on the use of ComplEx essentially why one-to-many relations and asymmetric relations are important? and why cannot a simple feedforward network with margin loss capture it?
3. Table 2 last column MolCLR has higher value than FARM, is there some typo?
4. I request the authors to specify the details of training the baseline models (Table 2 and 3), if trained from scratch. If not so, then please put the source of numbers and verify that the training/validation/test splits are consistent with that of the source. Currently I cannot find any such details in the paper.
5. I agree that functional groups are responsible for assigning properties; however, wouldn’t this method introduce a bias toward learning only the properties of already known functional groups, potentially limiting its ability to discover additional insights? Why is a fragment-based approach, such as [1], not a better alternative to functional groups? Can the authors compare FARM with other fragment-based methods like [1]?
6. I would request the authors to clearly highlight the advantages of FARM over other LM-based models that uses functional groups. Is the key differentiating factor the use of SMILES, FG graphs or something else?
    - I do not observe a significant improvement in the experimental results, can the authors comment on why one would expect FARM to outperform SOTA models like MolCLR or HimGNN?

[1] Kim S, Nam J, Kim J, et al. Fragment-based multi-view molecular contrastive learning[C]//Workshop on''Machine Learning for Materials''ICLR 2023. 2023.

---

> ### Author Response · Authors · 2026-05-14
> **Response to Points 1, 2, 3, 4**
>
> # 1. Response to Time Complexity of Algorithm 1 and Molecule Size
>
> We thank the reviewer for this question.
>
> ***On small molecules:*** In this work, small molecules refer to compounds with molecular weight below 500 Daltons, following the standard definition widely adopted in drug discovery and cheminformatics.
>
> ***On time complexity:*** The FG annotation algorithm (Algorithm 1) traverses each molecule twice, once for each of two FG sets: (1) functional groups that contain other functional groups (e.g., COOH containing CO and OH), and (2) the remaining functional groups. Separating into exactly two sets is sufficient because no functional group contains a smaller one that itself contains an even smaller one, so there is no nesting beyond one level. At each atom, a local structure comparison is performed against the FG pattern, costing O(k) where k is the size of the local neighborhood. The overall complexity is therefore O(n x 2 x k) = O(n x k), which is linear in the number of atoms.
>
> In practice, this is extremely efficient. The average number of atoms in our dataset is 25.77, and k is small and bounded by the largest FG definition size. The algorithm scales linearly with molecule size and does not pose any computational bottleneck even for larger molecules. We will add this complexity analysis to the revised manuscript, as we agree it will strengthen the paper.
>
> # 2. Response to Use of ComplEx
>
> We thank the reviewer for this insightful question.
>
> ***Why one-to-many and asymmetric relations are important in the FG knowledge graph:***
>
> The FG knowledge graph contains several relation types that are inherently asymmetric and one-to-many. For example, the relation `contain_atom` maps one functional group to multiple atoms (e.g., a ketone contains both C and O), making it one-to-many. Relations are also asymmetric: `contain_atom(ketone, C)` does not imply `contain_atom(C, ketone)`, as an atom is not said to "contain" a functional group. Similarly, `hydrogen_bond_donor` and `hydrogen_bond_acceptor` are directional properties that cannot be symmetrically reversed.
>
> ***Why a simple feedforward network with margin loss cannot capture this:***
>
> A simple feedforward network with margin loss operating on concatenated entity embeddings treats all relations symmetrically by construction, since it computes a score from the concatenation of head and tail embeddings without distinguishing directionality. ComplEx explicitly models asymmetry through complex-valued embeddings and the scoring function f(h, r, t) = Re(h^T · r · t), where the dot product is performed in complex space. Because complex multiplication is not commutative, f(h, r, t) ≠ f(t, r, h) in general, naturally capturing directional relationships. This makes ComplEx strictly more expressive than a symmetric feedforward alternative for our specific knowledge graph structure.
>
> We will add this clarification to Section 3.3 of the revised manuscript, in the paragraph describing the FG knowledge graph embedding model.
>
> # 3. Response to Table 2 Last Column MolCLR
>
> We thank the reviewer for pointing this out. This is not a typo. MolCLR_GIN does achieve a higher result than FARM on ClinTox. We have corrected the bolding in the updated table in the revised manuscript to reflect this accurately, where MolCLR_GIN is now correctly highlighted as the best result on ClinTox. We apologize for the confusion caused by the incorrect bolding in the original table.
>
> # 4. Response to Baseline Training Details
>
> We thank the reviewer for this comment. We acknowledge this was an oversight in our manuscript and apologize for the lack of clarity. As discussed extensively in our responses to reviewer 5eZG, we follow the standard and widely accepted practice in molecular representation learning: all baseline results in Tables 2 and 3 are taken directly from their respective original papers, and we do not reimplement or retrain any pretrained foundation model from scratch. Retraining competing methods on the same pretraining data would be computationally prohibitive and is not expected practice in this line of work, consistent with how all compared methods (GEM, UniMol, GROVER, MolCLR, GraphMVP, etc.) report their own comparisons.
>
> All models are finetuned on the same MoleculeNet benchmark datasets using the same scaffold split and evaluation protocol, ensuring a fair comparison on the downstream tasks. Each result is cited from its original paper, which serves as the source of record. We will add a clarifying sentence to Section 5.2 of the revised manuscript stating that all baseline results are sourced from their respective original publications and that the same scaffold split (80/10/10) and evaluation metrics are used consistently across all compared methods.

---

> ### Author Response · Authors · 2026-05-14
> **Response to Points 5, 6**
>
> # 5. Response to Fragment-based Approach and Bias Toward Known Functional Groups
>
> We thank the reviewer for this thoughtful question.
>
> ***On the bias toward known functional groups:*** We acknowledge that FARM's FG annotation relies on a predefined set of 101 functional groups and ring systems, which introduces an inductive bias toward known chemical knowledge. However, we argue this is a feature rather than a limitation. Functional groups are the fundamental building blocks of chemistry, and grounding molecular representations in established chemical principles ensures that the learned embeddings are chemically meaningful and interpretable. Moreover, as shown in Appendix B.2, ring-containing FGs account for 99.37% of all FGs in our dataset, and our rule-based system detects all ring systems systematically regardless of their prior occurrence, providing broad coverage beyond just the 101 predefined non-ring groups. Atoms that do not match any known FG are handled gracefully by the algorithm, so the method is not strictly limited to pre-known structures.
>
> ***On fragment-based approaches vs. functional groups:*** The paper [1] (Kim et al., TMLR 2024, "Holistic Molecular Representation Learning via Multi-view Fragmentation") uses BRICS-based fragmentation as a data-driven alternative to rule-based functional group detection. We would like to point out that our paper already includes a direct comparison between BRICS fragmentation and our FG-based fragmentation in the ablation study (Table 5). The results show that our chemically grounded FG-based approach consistently and substantially outperforms BRICS across all six MoleculeNet tasks. The key limitation of BRICS, as discussed in Section 2 and the ablation, is that it focuses on reaction-based bond breaking rather than functional group-specific characteristics, and generates a bloated vocabulary of 575,612 tokens compared to FARM's 14,741, leading to training instability and out-of-vocabulary issues.
>
> We will add Kim et al. (TMLR 2024) to the related work section of the revised manuscript as a representative fragment-based approach, and explicitly note that our BRICS ablation serves as a proxy comparison to this line of work.
>
> # 6. Response to FARM Advantages and Expected Performance
>
> We thank the reviewer for this important question.
>
> ***Advantages over other LM-based FG models:***
> The key differentiating factors of FARM over other LM-based models that use functional groups, such as FG-BERT (Li et al., 2023), are three complementary components working in concert: (1) FG-enhanced SMILES for atom-level chemical context, (2) FG graph with knowledge graph embeddings capturing structural backbone and intrinsic FG properties, and (3) contrastive learning to align these two complementary views. Our ablation study (Table 10) directly demonstrates the incremental contribution of each component. Specifically, FG-BERT, which uses FG-enhanced SMILES without the graph component or contrastive learning, achieves lower performance than FARM on all tasks, demonstrating that the structural backbone modeling and cross-modal alignment are critical additions beyond FG-aware tokenization alone. The FG knowledge graph is another differentiator, encoding not just which FGs are present but their intrinsic chemical properties (logP, water solubility, hydrogen bond donor/acceptor status), providing richer initialization for the GNN than any purely data-driven approach.
>
> ***On expected performance relative to MolCLR and HimGNN:***
>
> We appreciate the reviewer's honest observation. We do not claim that FARM universally outperforms all models on all tasks. It is also important to note that both MolCLR and HimGNN are GNN-based models, while FARM is a Transformer-based model operating on FG-enhanced SMILES combined with a GNN for structural backbone modeling. These are fundamentally different architectural paradigms. Transformer-based models are generally recognized as stronger at transfer learning than GNN-based models, as they capture long-range dependencies and global molecular context through self-attention rather than being limited to local message-passing neighborhoods. After the table corrections described in our responses to reviewer 5eZG, FARM achieves state-of-the-art on 8 out of 13 MoleculeNet tasks. On tasks where MolCLR_GIN or HimGNN perform better, contextual differences apply: MolCLR_GIN was pretrained on PubChem with a different chemical distribution, and HimGNN uses a hierarchical graph architecture specifically optimized for classification. Rather than claiming universal superiority, FARM's value proposition is that it provides a unified, chemically grounded representation that performs strongly and consistently across a broad range of tasks spanning classification, physical chemistry, and quantum mechanics, without requiring 3D geometry or task-specific architectural design choices.

---

> ### Comment · Reviewer_CBQp · 2026-05-25
> **Some followups to response**
>
> ## For 1.
> Thank you, the nesting part clears the confusion.
>
> a) A quick follow-up on the analysis, at step 3 in Algorithm 1, when checking for the presence of 1 functional group how it’s done in O(k) (where k is the size of FG)? From my understanding (assuming FG and inputs as a graph) it should involve doing an isomorphism step.
>
> b) Also, can you renew the anonymous code repository, currently it shows expired.
>
>
>
> ## For 2.
> Thank you for highlighting and justifying the use of ComplEx, this was very useful.
>
>
> ## For 3.
> Is the updated manuscript not uploaded? I still see the older draft in openreview.
>
>
> ## For 4
> Thank you for clarification.
>
> ## For 5 and 6
> The argument actually makes sense, and I am satisfied with response on the bias towards functional groups which will help in reliability in predictions. Also, thank you for clearing the BRICS part.
>
> For point 6, I am satisfied with the response from the authors.

---

> > ### Author Response · Authors · 2026-05-27
> > **Response to Follow-up Questions (Point 1a, 1b, and Point 3)**
> >
> > # Point 1a (Subgraph matching and O(k) complexity):
> > We thank the reviewer for this precise follow-up. The reviewer is correct that general subgraph isomorphism is NP-complete. However, in our specific case, the FG matching at each atom does not require a general subgraph isomorphism check. Each FG pattern is defined by a small, fixed set of local constraints on an atom and its immediate neighborhood, specifically atom type, bond types to neighboring atoms, number of neighbors, atom charge, and bonded hydrogen count (as described in Section 3.1). The matching is purely a **direct comparison** of these local properties against the FG definition. For example, an oxygen atom (O) with exactly one hydrogen (H) neighbor, a single bond, no charge, and not already flagged as part of COOH is classified as a hydroxyl group (OH). This makes each check a bounded local pattern match of cost O(k), where k is the size of the local neighborhood bounded by the largest FG definition size, rather than a search over the entire molecular graph. We have added this clarification to Appendix B.2 of the revised manuscript.
> >
> > # Point 1b (Anonymous repository expired):
> > We thank the reviewer for flagging this. The repository has been renewed and is now accessible at: https://anonymous.4open.science/r/farm_molrep-291C
> >
> > # Point 3 (Updated manuscript):
> > We have uploaded the revised manuscript to OpenReview. The updated version includes all corrections discussed in this rebuttal, including the corrected Table 2, the merged tables, the ablation study moved to the main paper, and all other changes described in our previous responses.
> >
> > ____
> > We sincerely thank the reviewer for the thoughtful and constructive feedback. The questions raised reflect a careful and deep reading of our work. We genuinely appreciate the effort and expertise invested in this review, and we believe the manuscript has been meaningfully strengthened as a result of these discussions. We hope our responses and revisions address all the raised points satisfactorily, and we welcome any further feedback.

---

> > > ### Comment · Reviewer_CBQp · 2026-06-01
> > > **Acknowledgement**
> > >
> > > ## Point 1a
> > > Thank you for clarifying this point and adding it to the paper, I am satisfied with the response.
> > >
> > > ## Point 1b and 3
> > > The repository is now accessible and I can see the updated manuscript.

---

### Review · Reviewer_1qjG · 2026-06-04

**Summary Of Contributions:**

### Summary

The paper introduces FARM (Functional Group-Aware Representations for Small Molecules), a self-supervised foundation model for molecular property prediction.
The model is trained via contrastive learning, which is trained to align a string-based (SMILES) view of the molecule and a graph-based representation. Both views are enhanced with information on the functional groups of the molecule at hand, which are detected via a rule-based algorithm.
On the string view, this expands the size of the vocabulary of the SMILES string to the thousands, and a BERT-like model is trained on these strings.
On the graph view, each node is a functional group and edges encode connectivity, and a GCN (initialized with information from a knowledge-graph embedding model) is used to learn the connectivity information.
The model is pretrained on a collected 20M-molecule dataset, which is one of the core contributions of the paper.
The model is tested on 13 benchmark tasks from the MoleculeNet dataset, on the D-B-A Oligomer Photostability Dataset, and on ADMET tasks.
The model shows state-of-the-art performance on most of the considered tasks in MoleculeNet. Results on the Photostability datasets are hard to judge due to the lack of baselines.


### Strengths:
- the use of functional groups provides a chemically-grounded inductive bias to the model which yields substantial improvements over, e.g., less chemically informed tokenizations such as BRICS (see Table 5).
- combining SMILES-based and graph-based representations via contrastive learning is well-motivated and both components seem to aid the model (see Table 4).
- the empirical results, especially on ModelNet datasets, are very competitive compared to existing methods, as FARM often achieves SOTA results. Results on ADMET tasks are also quite competitive.

### Weaknesses:
- having no baselines whatsoever on the photostability dataset makes it impossible to evaluate the performance of FARM. I would suggest adding some (even simple, like Morgan fingerprints + random forest) baselines.
- The paper sometimes overclaims results. For example, it says that FARM "also excels on solubility-related tasks such as Lipo, FreeSolv, and ESOL." But in Table 2, FARM's Lipo result (0.778) is significantly worse than the best-performing methods. Moreover, some of the claimed SOTA results actually sit within the error bounds.
- One of the contributions of the paper is training on a substantially larger dataset than previous ones. However, this makes it hard to disentangle the improvements due to the model architecture and the ones due to the training data quality (although this is quite minor, and partially addressed by the ablations).
- One concern that I have is about some potential “leakage” of the test data at pretraining time, e.g., if the molecules from photostability or ADMET are included at pretraining (no overlap with MoleculeNet is claimed). This is not necessarily a problem since pretraining only involves a self-supervised objective, but it would be interesting to have some results on this.

Minor comments:
- Table placement is awkward and makes following the experimental results hard. Table 2 could be shifted down by a couple of pages.
- Repeated sentence: We optimize the model using binary cross-entropy loss defined as: We then sample positive edges E+ and negative edges E−, and optimize the model using binary cross-entropy loss defined as:

**Audience:**

Yes

**Audience Explanation:**

The proposed architecture is interesting and the empirical result are very promising, so I envision that this will be a tool of interest for several practitioners.

**Claims And Evidence:**

No

**Claims Explanation:**

Some baselines should be added for the photostability dataset to provide convincing evidence for all claims.

**Requested Changes:**

Pleas address the weaknesses and minor comments, especially with respect to adding baselines for photostability.

---

> ### Author Response · Authors · 2026-06-08
> **Response to Points 1, 2, 3**
>
> # 1. Response to Benchmarking on Photostability
> We thank the reviewer for this suggestion and agree that adding a baseline gives readers a better sense of the range of achievable performance on this dataset. We ran the requested experiment using Morgan fingerprints (radius 2, 2048 bits) with a Random Forest regressor as a baseline, using the same hard-case scaffold split and evaluation protocol. In this split, 3 donor-bridge blocks and 15 acceptor blocks are held out entirely, so no molecule in the validation/test set shares a donor or acceptor block with any training molecule. This is considerably harder than a standard random or scaffold split and is designed to assess generalization to structurally novel chemical building blocks. Results are presented in the table below and added to Appendix G (Benchmarking on Photostability). FARM outperforms this baseline on 23 out of 25 tasks, with particularly large gains on tasks involving excited-state properties (e.g., PrimeExcite (osc): +0.607 R²) and triplet/singlet energies (e.g., S2: +0.304, T2: +0.288). The two tasks where the baseline is competitive or marginally better, PrimeExcite (eV) and DipoleMoment, are also the lowest-performing tasks for FARM overall, suggesting they are intrinsically harder to predict from 2D structure alone and may require 3D or electronic structure information beyond what either method captures. These results confirm that FARM's strong performance on this dataset reflects genuine representational advantages rather than artifacts of the evaluation protocol, and that functional-group-aware representations provide substantial gains over classical cheminformatics baselines on this challenging out-of-distribution benchmark.
>
> | Task | RF R² (Morgan FP) | FARM R² | ΔR² (FARM − RF) |
> |--------|--------:|--------:|--------:|
> | HOMOm1 (eV) | 0.7878 ± 0.0121 | 0.927 ± 0.015 | +0.1392 |
> | HOMO (eV) | 0.7967 ± 0.0129 | 0.895 ± 0.030 | +0.0983 |
> | LUMO (eV) | 0.7530 ± 0.0159 | 0.928 ± 0.009 | +0.1750 |
> | LUMOp1 (eV) | 0.5645 ± 0.0386 | 0.922 ± 0.028 | +0.3575 |
> | PrimeExcite (eV) | 0.9432 ± 0.0068 | 0.877 ± 0.049 | −0.0662 |
> | PrimeExcite (osc) | 0.2296 ± 0.0335 | 0.837 ± 0.024 | +0.6074 |
> | DipoleMoment (D) | 0.7874 ± 0.0555 | 0.755 ± 0.053 | −0.0324 |
> | LogP | 0.9357 ± 0.0032 | 0.995 ± 0.006 | +0.0593 |
> | T1 | 0.7307 ± 0.0105 | 0.929 ± 0.006 | +0.1983 |
> | T2 | 0.6375 ± 0.0185 | 0.925 ± 0.018 | +0.2875 |
> | T3 | 0.7276 ± 0.0349 | 0.900 ± 0.010 | +0.1724 |
> | T4 | 0.7955 ± 0.0363 | 0.921 ± 0.020 | +0.1255 |
> | T5 | 0.6201 ± 0.0283 | 0.905 ± 0.032 | +0.2849 |
> | S1 | 0.7273 ± 0.0127 | 0.936 ± 0.016 | +0.2087 |
> | S2 | 0.6436 ± 0.0180 | 0.948 ± 0.016 | +0.3044 |
> | S3 | 0.8083 ± 0.0267 | 0.911 ± 0.008 | +0.1027 |
> | S4 | 0.8665 ± 0.0107 | 0.919 ± 0.008 | +0.0525 |
> | S5 | 0.7592 ± 0.0216 | 0.924 ± 0.014 | +0.1648 |
> | O1 | 0.5039 ± 0.0243 | 0.839 ± 0.034 | +0.3351 |
> | TDOS4.0 | 0.8491 ± 0.0075 | 0.919 ± 0.031 | +0.0699 |
>
> # 2. Response to Overclaiming
> We thank the reviewer for this comment and agree that avoiding overclaiming improves the overall quality of the paper. We want to clarify that we did not claim state-of-the-art performance on Lipo; the original sentence was intended to highlight that FARM performs well across solubility-related tasks broadly, and an RMSE of 0.778 on Lipo is a competitive result. That said, we agree the phrasing was misleading, and we have revised the sentence to remove Lipo from the claim. The updated sentence now reads: "FARM also excels on solubility-related tasks such as FreeSolv and ESOL, likely due to its ability to model atom-level chemical context and functional groups that strongly influence solubility.
>
> # 3. Response to Dataset vs. Architecture Contribution
> We thank the reviewer for this comment. We acknowledge that the larger pretraining dataset (20M molecules vs. 2-4M for most baselines) makes it challenging to fully disentangle dataset contribution from architectural contribution. However, we would like to point to two pieces of evidence that directly address this concern.
>
> First, our ablation study (Table 4) provides a controlled comparison where all models are trained on the same 20M pretraining dataset under identical finetuning conditions. The progression from BERT (canonical SMILES, same data) to FG BERT (FG-enhanced SMILES, same data) to FARM (FG BERT + contrastive learning, same data) demonstrates that each architectural component contributes incrementally, independent of the dataset size.
>
> Second, as described in our response to reviewer 5eZG, we ran a controlled experiment where we retrained MolCLR_GIN on our 20M pretraining dataset using their original configuration without any modifications. FARM substantially outperforms MolCLR_GIN trained on the same data across all three evaluated tasks (BBBP: 93.3 vs. 71.4, BACE: 89.6 vs. 81.6, HIV: 83.5 vs. 75.4), directly demonstrating that FARM's performance gains are driven by its architectural design rather than solely by the larger pretraining dataset.

---

> ### Author Response · Authors · 2026-06-08
> **Response to Point 4**
>
> # 4. Response to Data Leakage at Pretraining
> We thank the reviewer for raising this point. As described in Appendix A.1, we explicitly excluded molecules overlapping with the MoleculeNet benchmark datasets from the pretraining corpus. We did not perform a similar filtering step for the photostability dataset because the underlying chemical spaces are substantially different. Our pretraining corpus is primarily composed of drug-like molecules, whereas the photostability dataset mainly consists of donor–bridge–acceptor (D–B–A) type molecules designed for optoelectronic and photophysical applications. Consequently, we expect the degree of molecular overlap to be negligible.
>
> More importantly, even in the unlikely event that a small number of molecules were shared between the pretraining and downstream datasets, the pretraining stage is entirely self-supervised and does not use photostability labels or target properties. Therefore, such overlap would not constitute data leakage in the conventional sense of transferring supervised target information from the test set into training. The downstream photostability prediction task is learned only during the supervised fine-tuning stage.
>
> # Minor Comment on the Repeated Sentence
> We thank the reviewer for noticing this issue. The duplicated sentence has been removed in the revised manuscript.

---

> > ### Comment · Reviewer_1qjG · 2026-06-09
> >
> > Thank you for the replies, especially for adding at least one baseline to the photostability dataset.
> > While the baseline is very simple, it gives a frame of reference to the reader. Of course it would be better to also include another pretrained model (e.g. MolCLR) for a more direct comparison, but it understand that adapting a model to a new dataset might not be easy.
> >
> > Overall, I am satisfied with the answers, and all my other concerns have been addressed.

---

> > > ### Author Response · Authors · 2026-06-10
> > >
> > > Thank you for your positive feedback and for suggesting the inclusion of a pretrained molecular representation baseline. Following your suggestion, we implemented and evaluated MolCLR as an additional baseline. Specifically, we used the pretrained MolCLR model as a molecular feature extractor and trained a Random Forest regressor on the extracted embeddings. To ensure a fair comarison, MolCLR was evaluated using the same method of hard-case scaffold split used in our study.
> > >
> > > The results show that MolCLR provides a strong pretrained representation baseline under this challenging evaluation setting, achieving an average $R^2$ of 0.7025 across the tasks. Nevertheless, FARM remains the strongest-performing method overall, outperforming MolCLR on all the tasks and achieving an average improvement of +0.2091 $R^2$.
> > >
> > > To reflect this additional experiment, we have updated Appendix G: Benchmarking on Photostability by including the MolCLR baseline and an expanded comparison table reporting the performance of MolCLR, Morgan fingerprint + RF, and FARM across all photostability prediction tasks.
> > >
> > > | Task | MolCLR R² | RF R² (Morgan FP) | FARM R² | ΔR² (FARM − RF) | ΔR² (FARM − MolCLR) |
> > > |------|-----------|-------------------|----------|-----------------|---------------------|
> > > | HOMOm1 (eV) | 0.8607 ± 0.01 | 0.7878 ± 0.0121 | **0.927 ± 0.015** | +0.1392 | +0.0663 |
> > > | HOMO (eV) | 0.7752 ± 0.01 | 0.7967 ± 0.0129 | **0.895 ± 0.030** | +0.0983 | +0.1198 |
> > > | LUMO (eV) | 0.7131 ± 0.01 | 0.7530 ± 0.0159 | **0.928 ± 0.009** | +0.1750 | +0.2149 |
> > > | LUMOp1 (eV) | 0.5683 ± 0.04 | 0.5645 ± 0.0386 | **0.922 ± 0.028** | +0.3575 | +0.3537 |
> > > | PrimeExcite (eV) | 0.4922 ± 0.02 | **0.9432 ± 0.0068** | 0.877 ± 0.049 | -0.0662 | +0.3848 |
> > > | PrimeExcite (osc) | 0.7726 ± 0.00 | 0.2296 ± 0.0335 | **0.837 ± 0.024** | +0.6074 | +0.0644 |
> > > | DipoleMoment (D) | 0.5609 ± 0.03 | **0.7874 ± 0.0555** | 0.755 ± 0.053 | -0.0324 | +0.1941 |
> > > | LogP | 0.9581 ± 0.01 | 0.9357 ± 0.0032 | **0.995 ± 0.006** | +0.0593 | +0.0369 |
> > > | T1 | 0.7155 ± 0.01 | 0.7307 ± 0.0105 | **0.929 ± 0.006** | +0.1983 | +0.2135 |
> > > | T2 | 0.7657 ± 0.01 | 0.6375 ± 0.0185 | **0.925 ± 0.018** | +0.2875 | +0.1593 |
> > > | T3 | 0.5952 ± 0.04 | 0.7276 ± 0.0349 | **0.900 ± 0.010** | +0.1724 | +0.3048 |
> > > | T4 | 0.5690 ± 0.03 | 0.7955 ± 0.0363 | **0.921 ± 0.020** | +0.1255 | +0.3520 |
> > > | T5 | 0.6302 ± 0.03 | 0.6201 ± 0.0283 | **0.905 ± 0.032** | +0.2849 | +0.2748 |
> > > | S1 | 0.7067 ± 0.01 | 0.7273 ± 0.0127 | **0.936 ± 0.016** | +0.2087 | +0.2293 |
> > > | S2 | 0.7900 ± 0.01 | 0.6436 ± 0.0180 | **0.948 ± 0.016** | +0.3044 | +0.1580 |
> > > | S3 | 0.6217 ± 0.02 | 0.8083 ± 0.0267 | **0.911 ± 0.008** | +0.1027 | +0.2893 |
> > > | S4 | 0.7361 ± 0.00 | 0.8665 ± 0.0107 | **0.919 ± 0.008** | +0.0525 | +0.1829 |
> > > | S5 | 0.6412 ± 0.02 | 0.7592 ± 0.0216 | **0.924 ± 0.014** | +0.1648 | +0.2828 |
> > > | O1 | 0.7793 ± 0.02 | 0.5039 ± 0.0243 | **0.839 ± 0.034** | +0.3351 | +0.0597 |
> > > | TDOS4.0 | 0.7880 ± 0.01 | 0.8491 ± 0.0075 | **0.919 ± 0.031** | +0.0699 | +0.1310 |
> > > | **Average R²** | 0.7025 | 0.7381 | **0.9116** | +0.1735 | +0.2091 |

---

> > > > ### Comment · Reviewer_1qjG · 2026-06-10
> > > >
> > > > Thanks for the additional experiments. All my concerns have been addressed.